# Improved Convergence of Differential Private SGD with Gradient Clipping

**Huang Fang, Xiaoyun Li, Chenglin Fan, Ping Li**

Cognitive Computing Lab
Baidu Research
No.10 Xibeiwang East Road, Beijing 100193, China
10900 NE 8th St. Bellevue, Washington 98004, USA
`{fangazq877,lixiaoyun996,fanchenglin,pingli98}@gmail.com`

## Abstract

Differential private stochastic gradient descent (DP-SGD) with gradient clipping (DP-SGD-GC) is an effective optimization algorithm that can train machine learning models with a privacy guarantee. Despite the popularity of DP-SGD-GC, its convergence in the unbounded domain without the Lipschitz continuous assumption is less-understood; existing analysis of DP-SGD-GC either impose additional assumptions or end up with a utility bound that involves a non-vanishing bias term. In this work, for smooth and unconstrained problems, we improve the current analysis and show that DP-SGD-GC can achieve a vanishing utility bound without any bias term. Furthermore, when the noise generated from subsampled gradients is light-tailed, we prove that DP-SGD-GC can achieve nearly the same utility bound as DP-SGD applies to the Lipschitz continuous objectives. As a by-product, we propose a new clipping technique, called value clipping, to mitigate the computational overhead caused by the classic gradient clipping. Experiments on standard benchmark datasets are conducted to support our analysis.

## 1 Introduction

Training machine learning models that can achieve decent prediction accuracy while preserving data privacy is fundamental in many modern machine learning applications. The concept of differential privacy (DP) from Dwork (2006); Dwork & Roth (2014) offers an elegant mathematical framework to characterize the privacy-preserving ability of randomized algorithms, which has been widely applied to tasks including clustering, regression, principle component analysis, empirical-risk minimization, matrix completion, graph distance estimation, optimization and deep learning (Chaudhuri & Monteleoni, 2008; Chaudhuri et al., 2011; Agarwal et al., 2018; Ge et al., 2018; Jain et al., 2018; Fan & Li, 2022; Fan et al., 2022). For the empirical-risk minimization (ERM) problem, among many proposed methods, differential private stochastic gradient descent (DP-SGD) is an effective algorithm that can solve the ERM problem with a privacy guarantee and achieve a reasonable utility bound. DP-SGD has received substantial interest in recent years due to its simplicity and effectiveness (Song et al., 2013; Bassily et al., 2014; Abadi et al., 2016; Wang et al., 2017; Bassily et al., 2019; Feldman et al., 2020; Asi et al., 2021).

In the classic analysis of DP-SGD, the variance of the Gaussian noise used in each iteration of DP-SGD relies crucially on the $\ell_2$-sensitivity of the loss function. Therefore most early works on DP-SGD assume each individual loss function to be Lipschitz continuous in its domain (Song et al., 2013; Bassily et al., 2014). However, many real-world problems are only smooth but not globally Lipschitz continuous; for example, the unconstrained linear regression problem. There are two techniques to circumvent the Lipschitz continuous assumption: (i) imposing an additional bounded domain constraint to the original problem; (ii) clipping gradients in their 2-norm and using the clipped gradients to update the model (Abadi et al., 2016). In practice, the gradient clipping technique is usually more preferred than imposing a bounded domain constraint because the latter requires prior knowledge of the distance between initialization and solution, which is typically unavailable for unconstrained problems. In summary, the state-of-the-art implementations of DP-SGD all advocate the gradient clipping technique.

Table 1: The utility bound and assumptions needed by different algorithms for convex problems, where $d$ is the problem size, $n$ is the number of data points and $\epsilon$ measures the privacy-preserving ability; see Section 3 for more details. "†" is based on a trivial extension of Bassily et al. (2014).

| Algorithms | Domain | Objective function | Light-tail-noise | Utility |
|---|---|---|---|---|
| DP-SGD (Bassily et al., 2014) | Bounded | Lipschitz cont. | No | $\widetilde{\mathcal{O}}(\sqrt{d}/(n\epsilon))$ |
| DP-SGD† | Bounded | Smooth | No | $\widetilde{\mathcal{O}}(\sqrt{d}/(n\epsilon))$ |
| DP-SGD (Wang et al., 2022) | Bounded | Hölder smooth | No | $\widetilde{\mathcal{O}}(\sqrt{d}/(n\epsilon))$ |
| DP-SGD-GC (Ours) | Unbounded | Smooth | Yes | $\widetilde{\mathcal{O}}(\sqrt{d}/(n\epsilon))$ |

Despite the popularity of DP-SGD with gradient clipping (DP-SGD-GC), the convergence of DP-SGD-GC for unconstrained problems that are not globally Lipschitz continuous has not been well-studied. In fact, recent works (Chen et al., 2020; Song et al., 2021) have reported that DP-SGD-GC can suffer from a constant utility in the worst case. With these negative results on the convergence of DP-SGD-GC, one may consider DP-SGD-GC as an algorithm with a fundamental non-convergence issue. In this work, we show that this is not the case. With a careful choice of the clipping threshold, we prove that DP-SGD-GC can achieve the same utility bound as its non-clipped counterpart DP-SGD. Formally, we summarize our contributions as follows.

- For unconstrained problems that are convex and smooth but not necessarily globally Lipschitz continuous, we show that DP-SGD-GC can achieve a $\widetilde{\mathcal{O}}(\sqrt{d}/(n\epsilon))$ utility bound when the noise generated from the subsampled gradients is light-tailed (Assumption 4.1)[1], which is the same as the utility bound of DP-SGD applies to the Lipschitz continuous problems. See Table 1 for a comparison to existing results. Our convergence analysis of DP-SGD-GC for convex, smooth and unconstrained problems, to our knowledge, provide the first utility bound without a non-vanishing bias term.

- We show that our analysis also applies to unconstrained smooth problems that can potentially be nonconvex. Consequently, DP-SGD-GC can achieve a $\widetilde{\mathcal{O}}(\sqrt{d}/(n\epsilon))$ gradient norm bound for smooth problems under the light-tail-noise assumption.

- This work is theoretical in essence but also includes a practical contribution (Section 5). We develop a novel *value clipping* technique for problems that satisfy the weak growth condition (Definition 3.1). The proposed value clipping technique can be implemented within one forward-backward propagation on existing learning platforms and can alleviate the computation overhead caused by gradient clipping. The efficiency of value clipping is demonstrated on real datasets.

## 2 RELATED WORK

DP-SGD with gradient clipping was initially proposed by Abadi et al. (2016). Gradient clipping and its variants have been widely adopted by many privacy-aware training algorithms (Andrew et al., 2021). Despite the popularity of gradient clipping, the convergence rate of DP-SGD-GC without the Lipschitz continuous and bounded domain assumptions remains a challenging task; see (Wang et al., 2022, Remark 5) for a short discussion on the hardness of removing the bounded domain assumption. This challenging research question was not carefully studied until the recent works from Chen et al. (2020) and Song et al. (2021), who provided counter-examples showing that DP-SGD-GC can suffer from a constant utility in the worst case. Chen et al. (2020) studied the convergence of DP-SGD-GC to a stationary point in the nonconvex setting and showed that an additional assumption on gradient distribution is sufficient to derive a meaningful utility bound. Song et al. (2021) showed that DP-SGD-GC converges to a perturbed objective function for the generalized linear model and can suffer from a constant utility for the original objective in the worst case. Note that there are some other recent works that study the convergence of DP-SGD-GC for smooth objective (Du et al., 2021; Wu et al., 2021; Yang et al., 2022), the rates in these works usually involve a bias term due to clipping. A concurrent work from Bu et al. (2022) suggests that a small clipping threshold can yield promising performance for DP-SGD-GC in certain scenarios, such as training language models. Their empirical discovery contrasts with the theoretical analysis in this work as our proof technique relies on a large clipping threshold. Bu et al. (2022)'s experiments indicate that the analysis in this

---

[1] The light-tail assumption is standard for deriving high probability error bound of SGD in the literature.

work may be further improvable; rigorous theoretical justification for the phenomenon described by Bu et al. (2022) is worth future investigation. Another concurrent work from Yang et al. (2022) studied the convergence of DP-SGD-GC under the generalized smooth condition, their analysis relies on a different set of assumptions and do not overlap with this work.

On the practical side, the original implementation of gradient clipping was inefficient as one needs to calculate the norm of each individual sample in every iteration. Many works (Goodfellow, 2015; Abadi et al., 2016; Rochette et al., 2019; Bu et al., 2021; Subramani et al., 2021) have been carried on to improve the efficiency of DP-SGD-GC from either engineering or algorithmic perspective. Our proposed value clipping technique can be viewed as an alternative to the classic gradient clipping.

## 3 PRELIMINARIES

**Notation** Throughout this paper, for any positive integer $n$, we denote $[n] := \{1, 2, \ldots, n\}$. We denote $\|\cdot\|$ to be the vector 2-norm or matrix operator norm if not otherwise specified. We use the notation $\widetilde{\mathcal{O}}(\cdot)$ to hide poly-logarithmic terms.

We consider the empirical-risk minimization (ERM) problem

$$\underset{w \in \mathbb{R}^d}{\text{minimize}} \quad f(w) := \frac{1}{n} \sum_{i=1}^{n} f_i(w), \tag{P}$$

where $n$ is the number of training samples, $f_i$'s are differentiable functions and $w$ is the model we wish to train. Throughout the paper, we assume that $f$ is bounded below and its minimum is attainable. We let $\mathcal{W}^*$ to be the set of solutions for problem (P), and denote the optimal function value as $f^*$. We assume that a lower bound of $f^*$ is known as a prior. Note that this assumption holds in many realistic settings, for example the ERM problems are usually lower bounded by zero.

Next we introduce the *weak growth condition* (WGC), which is the cornerstone for our analysis.

**Definition 3.1.** A function $h : \mathbb{R}^d \to \mathbb{R}$ is $(\beta_1, \beta_2)$-WGC for some $\beta_1 > 0, \beta_2 \geq 0$ if

$$\|\nabla h(w)\|^2 \leq \beta_1 \Big( h(w) - \inf_{u \in \mathbb{R}^d} h(u) \Big) + \beta_2 \qquad \forall w \in \mathbb{R}^d.$$

The weak growth condition bounds the norm of the gradient by a linear function of the objective value. WGC is gaining increasing interest in recent years as multiple works have demonstrated that WGC and its variants can improve the classic analysis of SGD-type algorithms (Schmidt & Roux, 2013; Needell et al., 2014; Vaswani et al., 2019; Qian et al., 2019; Stich, 2019; Khaled & Richtárik, 2020; Fang et al., 2021; Gower et al., 2021). It is easy to show that smooth functions that are bounded below necessarily satisfy WGC; the following Lemma makes this precise.

**Lemma 3.2.** *If a function $h : \mathbb{R}^d \to \mathbb{R}$ is $L$-smooth for some $L > 0$ and bounded below, e.g., $\inf_{w \in \mathbb{R}^d} h(w) > -\infty$. Then $h$ is $(2L, 0)$-WGC.*

We stress that WGC is not a strong assumption. In fact, a wide range of nonconvex and nonsmooth problems arising from the ERM problem also satisfy WGC, e.g., the Lipschitz continuous model with smooth and convex loss function; see Fang et al. (2021, § 4.1) and Section D for more details.

We recall the standard definition of differential privacy (DP).

**Definition 3.3** (Dwork, 2006). A randomized algorithm $\mathcal{A}$ is $(\epsilon, \delta)$-differentially private if for all neighboring datasets $D, D'$ and for all events $S$ in the output space of $\mathcal{A}$, we have

$$\Pr[\mathcal{A}(D) \in S] \leq e^\epsilon \Pr[\mathcal{A}(D') \in S] + \delta.$$

If $\delta = 0$, then $\mathcal{A}$ is said to be $\epsilon$-differentially private.

The detailed algorithm of DP-SGD-GC is shown in Algorithm 1. It has been shown that DP-SGD-GC is $(\epsilon, \delta)$-DP as long as the noise level $\sigma$ is larger than certain threshold (Theorem 3.4).

**Theorem 3.4** (Abadi et al., 2016, Theorem 1). *Let $q = B/n$, where $B$ is the batch size and $n$ is the number of data points. There exist constants $c_1$ and $c_2$, such that for any $\epsilon < c_1 q^2 T$, Algorithm 1 is $(\epsilon, \delta)$-DP for any $\delta > 0$ if $\sigma \geq c_2 \frac{q\sqrt{T \log(1/\delta)}}{\epsilon}$.*

---

**Algorithm 1** Differential-private SGD with gradient clipping (DP-SGD-GC)

---

1: **Input:** number of iteration $T \in \mathbb{N}$, clipping threshold $C > 0$, noise level $\sigma > 0$, batch size $B \in [1, n]$, learning rate $\eta > 0$, initial iterate $w^{(0)}$.
2: **for** $t \leftarrow 0, \ldots, T - 1$ **do**
3:      Sample a mini-batch $\mathcal{B}_t$, where each data has probability $B/n$ to be sampled;
4:      $g_i^{(t)} = \nabla f_i(w^{(t)}) \quad \forall i \in \mathcal{B}_t$;
5:      $\tilde{g}_i^{(t)} = g_i^{(t)} / \max\{1, \|g_i^{(t)}\|/C\}, \quad \forall i \in \mathcal{B}_t$;
6:      $w^{(t+1)} = w^{(t)} - \eta \frac{1}{B} \left( \sum_{i \in \mathcal{B}_t} \tilde{g}_i^{(t)} + \xi^{(t)} \right), \quad$ where $\xi^{(t)} \sim \mathcal{N}(0, C^2 \sigma^2 \mathbf{I}_{d \times d})$;
7: **end for**
8: **Return:** $w^{(i)}$ where $i$ is uniform randomly sampled from $\{0, 1 \ldots, T\}$.

---

## 4   MAIN THEORETICAL RESULTS

We present our main theoretical contributions in this section. Denote $w_{\text{priv}}$ as the output of Algorithm 1. We are interested in the upper bound of the *excess empirical risk* $f(w_{\text{priv}}) - f^*$ and the *gradient norm square* $\|\nabla f(w_{\text{priv}})\|^2$ without assuming $f_i$'s to be Lipschitz continuous in $\mathbb{R}^d$. Part of our analysis relies on assuming the noise generated from subsampled gradients is "light-tailed" (sub-Gaussian). Formally, we introduce the following assumption[2].

**Assumption 4.1.** *There exist $\rho > 0$ such that*
$$\mathbb{E}_i \left[ \exp \left( \|\nabla f_i(w) - \nabla f(w)\|^2 / \rho^2 \right) \right] \leq e, \qquad \forall w \in \mathbb{R}^d.$$

We note that the above light-tail assumption is a widely used assumption for the analysis of high probability utility bound of SGD (Nemirovskii et al., 2009; Juditsky & Nesterov, 2014; Ghadimi & Lan, 2013; Harvey et al., 2019; Feldman et al., 2020). Assumption 4.1 does not imply $f_i$'s to be globally Lipschitz continuous on $\mathbb{R}^d$, and therefore will not trivialize our analysis. Note that there is a recent trend on analyzing SGD and its variants with heavy-tail gradient noise (Gürbüzbalaban et al., 2021). Our analysis does not apply to the heady-tail setting because our Proposition 4.2 relies crucially on the light-tail-noise assumption.

The idea of our proof is concise. We first summarize the proof sketch as follows, and then explain the technical details step by step.

---

**Proof sketch:**

- Assuming that the initial objective gap $f(w^{(0)}) - f^*$ is bounded. We are able to prove that, with high probability, the iterates generated from SGD have bounded objective values that only logarithmically depend on $T$.

- The weak growth condition allows us to convert the objective value upper bound to the gradient norm upper bound. Thus, with high probability, gradient clipping will never happen during the process of DP-SGD if the clipping threshold $C$ is chosen appropriately.

- Finally, we can apply the classic convergence analysis of SGD (without gradient clipping) and obtain a non-trivial excess empirical risk or gradient norm upper bound.

---

To begin with, we develop a uniform upper bound on the objective values, e.g., $f(w^{(t)})$ where $w^{(t)}$'s are the iterates generated from the vanilla SGD algorithm (Algorithm 2) with sub-Gaussian noise.

**Proposition 4.2** (Uniform upper bound on objective values of SGD with sub-Gaussian noise). *Assume $f$ is $L$-smooth for some $L > 0$ and there exist $\tilde{\sigma} > 0$ such that $\mathbb{E}[\exp(\|\zeta^{(t)}\|^2 / \tilde{\sigma}^2)] \leq e$ for any $t \in \mathbb{N}$. Denote $\{w^{(t)}\}_{t=0}^{T}$ as the iterates generated from Algorithm 2 with $\eta \leq \min \left\{ \frac{1}{2L}, \frac{1}{\tilde{\sigma}\sqrt{T}} \right\}$. Then for any $\delta \in (0, 1)$,*
$$\max_{t \in \{0, 1, \ldots, T\}} \left( f(w^{(t)}) - f^* \right) \leq 2 \left( f(w^{(0)}) - f^* \right) + \mathcal{O} \left( \log(T/\delta) \right)$$
*with probability at least $1 - \delta$.*

---

[2]Note that there are several equivalent (up to constant) definitions of sub-Gaussian variable; see Vershynin, 2018, Proposition 2.5.2. These definitions are often used interchangeably in the literature.

---

**Algorithm 2** Stochastic gradient descent

---

1: **Input:** total iterations $T \in \mathbb{N}$, learning rate $\eta > 0$, initial iterate $w^{(0)}$.
2: **for** $t \leftarrow 0, \ldots, T-1$ **do**
3: $\quad w^{(t+1)} = w^{(t)} - \eta \left( \nabla f(w^{(t)}) + \zeta^{(t)} \right);$
4: **end for**
5: **Return:** $w^{(i)}$ where $i$ uniform randomly sampled from $\{0, 1 \ldots, T\}$.

---

The conclusion stated in Proposition 4.2 may seem obvious at first glance as the training algorithm is expected to produce almost decreasing objective values during the optimization process. However, we note that deriving a nearly constant upper bound of $f(w^{(t)}) - f^*$ that holds uniformly over $t \in \{0, 1 \ldots, T\}$ without assuming Lipschitz continuity or bounded domain is nontrivial. Our proof relies on a recently proposed technical tool called the generalized Freedman inequality (Harvey et al., 2019); see Section B.1 for the detailed proof of Proposition 4.2. We also remark that Proposition 4.2 holds for the standard SGD algorithm without considering differential privacy, and thus may be of independent interest.

Based on Proposition 4.2, we can further obtain an upper bound on each individual loss $f_i(w^{(t)}) - f_i^*$ under Assumption 4.1, and therefore also upper bound $\|\nabla f_i(w^{(t)})\|$ via the weak growth condition.

**Proposition 4.3.** *Assume $f_i$'s are $L$-smooth for some $L > 0$ and there exist $\tilde{\sigma} > 0$ such that $\mathbb{E}[\exp(\|\zeta^{(t)}\|^2 / \tilde{\sigma}^2)] \le e$ for any $t \in \mathbb{N}$. Denote $\{w^{(t)}\}_{t=0}^T$ as the iterates generated from Algorithm 2 with $\eta \le \min\left\{ \frac{1}{2L}, \frac{1}{\tilde{\sigma}\sqrt{T}} \right\}$. Then for any $\delta \in (0,1)$,*

$$\max_{i \in [n], t \in [T]} \|\nabla f_i(w^{(t)})\| \le \sqrt{2\beta_1(f(w^{(0)}) - f^*)} + \mathcal{O}\left( \sqrt{\log(1/\delta)} + \sqrt{\log T} + \sqrt{\log n} \right) \quad (1)$$

*holds with probability at least $1 - \delta$ for any $\delta \in (0,1)$.*

When Assumption 4.1 holds, Proposition 4.3 suggests that the upper bound of the maximum gradient norm logarithmically depends on $\delta, T$ and $n$ (eq. (1)).

Now we are ready to present our excess empirical risk bound and gradient norm bound for DP-SGD-GC in terms of conditional expectation.

**Proposition 4.4** (Convergence on conditional expectation). *Assume that Assumption 4.1 holds. Denote $w_{\mathrm{priv}}$ as the output of Algorithm 1 and define $\mathcal{E} := \{\|f_i(w^{(t)})\| \le C \, \forall i \in [n], t \in \{0, 1 \ldots, T\}\}$ as the event of no clipping happens during the training of Algorithm 1. Let $D_f := f(w^{(0)}) - f^*$. Given any $\epsilon > 0$ and $\delta, \delta' \in (0.5, 1)$.*

- *Assume $f_i$'s are convex and $L$-smooth for some $L > 0$. set $T > \epsilon/c_1$,*

$$\sigma = \frac{c_2 B \sqrt{T \log(1/\delta)}}{n\epsilon}, \ C = \sqrt{c_3 + c_4 \log(nT/\delta')}, \ \eta = \min\left\{ \frac{1}{2L}, \frac{c_5 B}{\sqrt{(B\rho^2 + C^2 d\sigma^2)T}} \right\}, \quad (2)$$

*where $c_1, c_2, c_5$ are some absolute constants and $c_3, c_4$ are constants that depend on $L$ and $D_f$. We have that Algorithm 1 is $(\epsilon, \delta)$-DP, $\Pr[\mathcal{E}] \ge 1 - \delta'$, and*

$$\mathbb{E}\left[ f(w_{\mathrm{priv}}) - f^* \mid \mathcal{E} \right] \le \mathcal{O}\left( \frac{1}{T} + \frac{1}{\sqrt{BT}} + \frac{\left( \sqrt{\log(1/\delta')} + \sqrt{\log(Tn)} \right) \sqrt{d\log(1/\delta)}}{n\epsilon} \right).$$

- *Assume $f_i$'s are $L$-smooth for some $L > 0$. Setting $\epsilon, \sigma, C, \eta$ as in eq. (2). It holds that Algorithm 1 is $(\epsilon, \delta)$-DP, $\Pr[\mathcal{E}] \ge 1 - \delta'$, and*

$$\mathbb{E}\left[ \|\nabla f(w_{\mathrm{priv}})\|^2 \mid \mathcal{E} \right] \le \mathcal{O}\left( \frac{1}{T} + \frac{1}{\sqrt{BT}} + \frac{\left( \sqrt{\log(1/\delta')} + \sqrt{\log(Tn)} \right) \sqrt{d\log(1/\delta)}}{n\epsilon} \right).$$

While the bounds stated in Proposition 4.4 are close to our objective, we note that these bounds are expressed in terms of conditional expectation where the conditioning event happens with high probability, which is different from the classic notion of convergence in expectation. Fortunately, we show that it is easy to convert the bounds in Proposition 4.4 to expected utility bound when the random variable $f(w_{\mathrm{priv}}) - f^*$ (or $\|\nabla f(w_{\mathrm{priv}})\|^2$) is sub-exponential, which is true under Assumption 4.1. Lemma A.9 serves as the main technical tool for this conversion.

**Theorem 4.5** (Convergence on expectation). *Suppose that Assumption 4.1 holds. Denote $w_{\mathrm{priv}}$ as the output of Algorithm 1. Let $D_f \coloneqq f(w^{(0)}) - f^*$. Given any $\epsilon > 0$ and $\delta \in (0.5, 1)$. Setting $T, \sigma, \eta$ in the same way as eq. (2) and let $C = \sqrt{c_3 + c_4 \log(nT)}$, where $c_1, c_2, c_5$ are some absolute constants and $c_3, c_4$ are constants that depend on $L, D_f$.*

- *Assume $f_i$'s are convex and $L$-smooth for some $L > 0$. Then Algorithm 1 is $(\epsilon, \delta)$-DP and*

$$\mathbb{E}\left[f(w_{\mathrm{priv}}) - f^*\right] \leq \widetilde{\mathcal{O}}\left(\frac{1}{T} + \frac{1}{\sqrt{BT}} + \frac{\sqrt{d}}{n\epsilon}\right).$$

*Consequently, we have $\mathbb{E}\left[f(w_{\mathrm{priv}}) - f^*\right] = \widetilde{\mathcal{O}}\left(d^{1/2}(n\epsilon)^{-1}\right)$ by setting $T = \Theta\left(n^2\epsilon^2 d^{-1}\right)$.*

- *Assume $f_i$'s are $L$-smooth for some $L > 0$. Then Algorithm 1 is $(\epsilon, \delta)$-DP and*

$$\mathbb{E}\left[\|\nabla f(w_{\mathrm{priv}})\|^2\right] \leq \widetilde{\mathcal{O}}\left(\frac{1}{T} + \frac{1}{\sqrt{BT}} + \frac{\sqrt{d}}{n\epsilon}\right).$$

*Consequently, we have $\mathbb{E}\left[\|\nabla f(w_{\mathrm{priv}})\|^2\right] = \widetilde{\mathcal{O}}\left(d^{1/2}(n\epsilon)^{-1}\right)$ by setting $T = \Theta\left(n^2\epsilon^2 d^{-1}\right)$.*

**Remark 4.1.** *Our results suggest that, when the problem is smooth, the Lipschitz continuous and bounded domain assumptions can be removed almost for free when analyzing DP-SGD-GC with light-tailed gradient noise; the only cost is some logarithmic terms.*

**Remark 4.2.** *Our analysis also holds for DP-GD-GC. When analyzing DP-GD-GC, Assumption 4.1 is no longer required. However, when Assumption 4.1 does not hold, there will be an additional multiplicative term $\sqrt{n}$ appear in eq. (1) and the final utility bound is $\widetilde{\mathcal{O}}(\sqrt{d}/(\sqrt{n}\epsilon))$.*

Theorem 4.5 is the main theoretical contributions of this paper. The bounds stated in Theorem 4.5 nearly match the rate $\mathcal{O}(\log(1/\delta)d^{1/2}(n\epsilon)^{-1})$, which is the best-known bound of DP-SGD with the Lipschitz continuous or bounded domain assumption (Bassily et al., 2014). To our knowledge, for unconstrained smooth problems, this is the first utility bound of DP-SGD-GC without a non-vanishing bias term. Note that existing lower bound analyses for DP-SGD either assume the domain is bounded (Bassily et al., 2014) or the loss is Lipschitz continuous (Song et al., 2021). Therefore those lower bounds are not comparable with the upper bounds stated in Theorem 4.5. We leave the lower bound of DP-SGD-GC for unconstrained smooth problems as a future direction to explore.

## 5 VALUE CLIPPING

This section focuses on the practical side of DP-SGD-GC. A well-known implementation issue of Algorithm 1 is that the gradient clipping step (line 4 of Algorithm 1) requires to access the norm of each individual gradient from the sampled batch, and naive implementation of DP-SGD-GC on current deep learning platforms cannot fully exploit the parallelism of GPU; see some attempts that try to mitigate this issue (Goodfellow, 2015; Rochette et al., 2019; Bu et al., 2021). The state-of-the-art implementation of DP-SGD-GC is from Subramani et al. (2021), who developed a highly engineered approach to exploit language primitives, compilation, and vectorization on certain deep learning platforms. In this section, we propose a *value clipping* technique for functions that satisfy the weak growth condition (Definition 3.1). The proposed value clipping can be viewed as an alternative to the classic gradient clipping technique that is easy to implement on all existing deep learning platforms such as PaddlePaddle.

The intuition behind value clipping is simple — when $f_i$'s satisfy the weak growth condition, the norm of their gradients can be bounded by a function of their objective values, e.g., $\|\nabla f_i(w)\| \leq \sqrt{\beta_1(f_i(w) - f_i^*) + \beta_2}$, therefore scaling the gradient by $\sqrt{\beta_1(f_i(w) - f_i^*) + \beta_2}$ ensures the scaled gradient has a bounded norm. Formally,

---

**Algorithm 3** Differential-private SGD with value clipping (DP-SGD-VC)

---

1: **Input:** number of iteration $T \in \mathbb{N}$, clipping threshold $C > 0$, noise level $\sigma > 0$, batch size $B \in [1, n]$, learning rate $\eta > 0$, initial iterate $w^{(0)}$, WGC parameters $\beta_1 > 0, \beta_2 \geq 0, f_{\text{lb}}^* \in \mathbb{R}$ that lower bound $f_i^* \ \forall i \in [n]$.
2: **for** $t \leftarrow 0, \ldots, T-1$ **do**
3:     Sample a mini-batch $\mathcal{B}_t$, where each data have probability $B/n$ to be sampled;
4:     $\tilde{g}_i^{(t)} = \nabla f_i(w^{(t)}) / \max \left\{ 1, \sqrt{\beta_1(f(w^{(t)}) - f_{\text{lb}}^*) + \beta_2}/C \right\} \qquad \forall i \in \mathcal{B}_t$;
5:     $w^{(t+1)} = w^{(t)} - \eta \frac{1}{B} \left( \sum_{i \in \mathcal{B}_t} \tilde{g}_i^{(t)} + \xi^{(t)} \right), \quad$ where $\xi^{(t)} \sim \mathcal{N}(0, C^2\sigma^2 \mathbf{I}_{d \times d})$;
6: **end for**
7: **Return:** $w^{(i)}$ where $i$ uniform randomly sampled from $\{0, 1 \ldots, T\}$.

---

$$\forall w \in \mathbb{R}^d, \quad \tilde{g} := f_i(w) / \max \left\{ 1, \frac{\sqrt{\beta_1(f_i(w) - f_i^*) + \beta_2}}{C} \right\} \implies \|\tilde{g}\| \leq C.$$

The detailed algorithm of DP-SGD with value clipping, termed DP-SGD-VC, is shown in Algorithm 3. Note that DP-SGD-VC requires knowing the WGC parameters $\beta_1, \beta_2$ and a lower bound of $f_i^*$'s as its input. The WGC parameters for simple models, including linear and logistic regression, are easy to calculate. For feed-forward neural networks, the calculation of WGC parameters is achievable but more involved; see Appendix D for details. It is easy to verify that DP-SGD-VC is $(\epsilon, \delta)$-DP because the norm of the clipped gradient is guaranteed to be bounded by $C$; the following corollary is a direct consequence of Theorem 3.4 and describes the DP property of DP-SGD-VC.

**Corollary 5.1.** *Assume $f_i$'s are $(\beta_1, \beta_2)$-WGC for some $\beta_1 > 0, \beta_2 \geq 0$. Let $q = B/n$, where $B$ is the batch size and $n$ is the number of data points. There exist constants $c_1$ and $c_2$, such that for any $\epsilon < c_1 q^2 T$, Algorithm 3 is $(\epsilon, \delta)$-DP for any $\delta > 0$ if $\sigma \geq c_2 q \sqrt{T \log(1/\delta)} \epsilon^{-1}$.*

**Remark 5.1.** *DP-SGD-VC is easy to implement on existing auto-differentiation based deep learning platforms. The value clipping step (line 4 of Algorithm 3) can be realized within one forward-backward propagation if the WGC parameters are given in advance. Therefore DP-SGD-VC can be as fast as the vanilla SGD algorithm.*

## 6 NUMERICAL STUDY

We conduct experiments on two standard image classification benchmark datasets: MNIST (LeCun, 1998) and CIFAR10 (Krizhevsky & Hinton, 2009). In Appendix, we also present some experimental results on synthetic data with light-tailed noise. For MNIST, we train a linear classifier and a two-layer MLP with 128 hidden nodes respectively. For CIFAR10, to achieve decent accuracy, we use a pre-trained VGG16 network (Simonyan & Zisserman, 2015) to extract informative high-level features. Based on the 512-dimensional extracted features, we train a linear classifier and a two-layer MLP with 128 hidden nodes respectively.

**Implementation details** For all experiments, we set the batch size $B = 128$, the noise level $\sigma = 1.0$ and the confidence level $\delta = 10^{-5}$. For MNIST, we try learning rate in $\{2 \times 10^{-3}, 5 \times 10^{-3}, 10^{-2}\}$ for each experiment and report the best result. For CIFAR10 we fix the learning rate to be $0.1$. All experiments are conducted on a server with 4 CPUs and one NVIDIA Tesla P100 GPU.

### 6.1 THE EVOLUTION OF CLIPPING FREQUENCY DURING TRAINING

We run DP-SGD-GC with different clipping thresholds. In particular, we try $C \in \{1, 5, 20, 40\}$ and $C \in \{0.1, 0.2, 0.4, 1.0\}$ for MNIST and CIFAR10 respectively. The evolutions of training accuracy and clipping frequency per epoch with different clipping thresholds are shown in Figure 1. We can observe that, in most cases, the clipping frequency decreases as the training accuracy goes up. This observation aligns with the WGC as lower training loss implies a smaller average gradient norm and further results in lower clipping frequency. We also see that, in most cases, the clipping frequency can become close to 0 when the clipping threshold is chosen appropriately; this observation is consistent with our theoretical analysis. Another interesting observation is the result of training the two-layer neural network with MNIST and $C \in \{20, 40\}$. The clipping frequency is small initially and becomes stable at 15% instead of decreasing to zero. We conjecture that this phenomenon is because the WGC

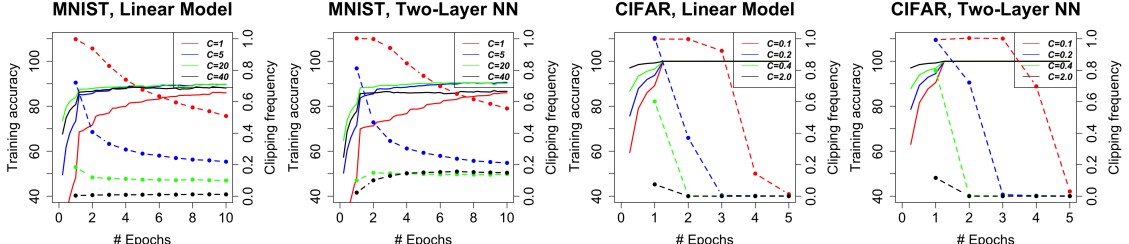

Figure 1: The evolution of training accuracy and clipping frequency during DP-SGD-GC, where the solid line ▬ represent training accuracy and the dashed line ▪ ▪ ▪ denote the clipping frequency per epoch. The left two figures present the results on the MNIST dataset, the algorithm is about $(1.0379, 10^{-5})$-DP. The right two figures show the results on the CIFAR10 dataset (feature extracted by a pretrained VGG16 network), the algorithm is about $(0.9580, 10^{-5})$-DP.

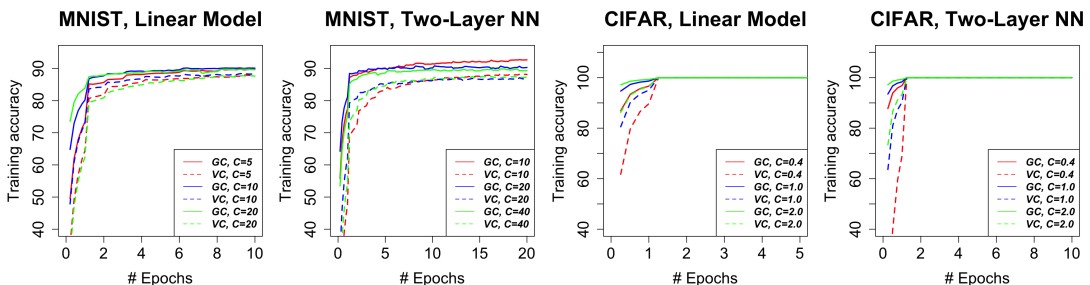

Figure 2: Value clipping (VC) versus gradient clipping (GC) in terms of training accuracy.

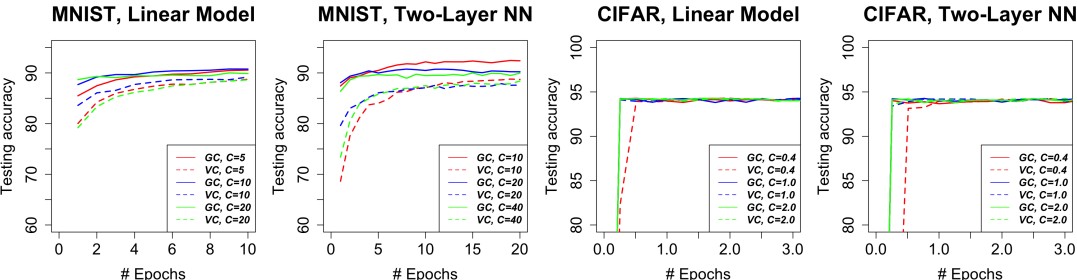

Figure 3: Value clipping (VC) versus gradient clipping (GC) in terms of testing accuracy.

parameters of the neural network grow as the training goes and thus prevent some gradient norms from being smaller than the clipping threshold.

## 6.2 THE EVALUATION OF DP-SGD-VC

We set $f_{1b}^* = 0$ for all experiments with DP-SGD-VC. The calculation of the WGC parameters for a feed-forward neural network with cross-entropy loss is given in Section D, where $\beta_2 = 0$ and $\beta_1$ depends on the spectral norm of each layer of neural networks. Compared with vanilla SGD, DP-SGD-VC has an additional cost to calculate the spectral norm of each layer in each iteration. As shown in the following content, the overhead of calculating the spectral norm is not significant.

**Training and testing accuracy** The training and testing accuracy of DP-SGD-GC and DP-SGD-VC with different clipping thresholds are shown in Figure 2 and Figure 3. We can observe that DP-SGD-VC converges slightly slower than DP-SGD-GC in terms of the epoch. This observation should not be surprising as DP-SGD-VC uses an upper bound of the gradient norm for clipping and will result in a smaller effective learning rate than DP-SGD-GC. For CIFAR10, the training and testing are easy as the model is trained on pre-trained features; both DP-SGD-VC and DP-SGD-GC can achieve similar accuracy at the end of training. For MNIST, there is an unfortunate loss of training and testing accuracy. For MNIST with the linear model, there is a $\sim 2\%$ loss in training and testing

Table 2: Per epoch runtime of different methods. SGD without gradient clipping is the baseline method; Micro-batching is the naive implementation of DP-SGD-GC; GC and VC are the classic gradient clipping and the proposed value clipping, respectively. We bold the shortest per epoch runtime among methods besides SGD.

| Data | Model | SGD | Micro-batching | GC-Opacus | DP-SGD-VC |
|------|-------|-----|----------------|-----------|-----------|
| MNIST | Linear | 6.83s | 34.82s | 12.80s | **8.62**s |
| MNIST | 2-layer NN | 6.91s | 43.75s | 16.61s | **8.96**s |
| CIFAR | Linear | 0.58s | 23.99s | 4.13s | **1.25**s |
| CIFAR | 2-layer NN | 0.65s | 30.86s | 7.47s | **1.70**s |

accuracy. For MNIST with two-layer NN, there is a $2 \sim 3\%$ loss in training and testing accuracy for $C \in \{20, 40\}$ and a $\sim 4\%$ loss for $C = 10$. The gap between DP-SGD-GC and GP-SGD-VC is more obvious when the clipping threshold is small. We conjecture that the loss of training and testing accuracy is due to our estimation of $f_i^*$. The estimation $f_{1b}^* = 0$ is accurate for CIFAR10 as the model can almost perfectly fit all pre-trained data. However, the estimation is inaccurate for MNIST and thus results in a loose upper bound of the gradients' norm.

**Comparing the computational time per epoch** We report the per epoch runtime of different algorithms in Table 2. All experiments are conducted on a server with one NVIDIA Tesla P100 GPU. The vanilla SGD without privacy consideration is the baseline method and is the fastest among all algorithms. Micro-batching is the naive implementation of DP-SGD-GC and is significantly slower than SGD. GC-Opacus is the implementation of DP-SGD-GC from the (highly optimized) Opacus package; we can see that there is still a gap between the performance of GC-Opacus and the standard non-private SGD algorithm. DP-SGD-VC is our implementation of DP-SGD with the proposed value clipping technique. We can observe that DP-SGD-VC is slightly slower than the standard SGD algorithm and faster than other private training methods.

**Limitations** Despite the efficiency of DP-SGD-VC, it also has certain limitations: (i) as shown in the experiments, DP-SGD-VC may cause some loss in training/testing accuracy if our estimation for the WGC parameters is loose; (ii) DP-SGD-VC requires calculating the WGC parameters, which we show is available for feed-forward neural works with cross-entropy loss. However, it would be hard to apply VC to more complicated network architectures with arbitrary loss in a black-box manner, e.g., transformers with ranking loss. Overall, we consider DP-SGD-VC as an alternative for DP-SGD-GC that is computationally cheap; DP-SGD-VC can perform similarly to DP-SGD-GC in certain scenarios but can also be not applicable in other situations.

## 7  CONCLUSION AND FUTURE WORK

This paper studied the convergence behavior of a widely used privacy-preserving learning algorithm called DP-SGD-GC. Our analysis extended the convergence of DP-SGD-GC to smooth and unconstrained problems without assuming the objective to be globally Lipschitz continuous. We believe that our theoretical results improved the current understanding of DP-SGD-GC and provided new insights for practitioners and researchers to use DP-SGD-GC and design new algorithms. Our analysis can potentially be used for other privacy-preserving learning algorithms such as adaptive DP-SGD (Asi et al., 2021) and DP-SGD with subspace identification (Zhou et al., 2021; Song et al., 2021).

Our work implies some future directions. Firstly, the light-tail noise condition may not hold in some machine learning applications (Gürbüzbalaban et al., 2021). In these scenarios, the utility bound of DP-GD-GC (Remark 4.2) is $\sqrt{n}$ worse than the best-known bound of DP-SGD-GC for Lipschitz functions. Whether it is possible to further improve the utility bound with heavy-tail-noise is an interesting question. Another direction is to explore the lower bound of DP-SGD-GC with a carefully tuned clipping threshold for smooth and unconstrained problems. Lastly, we may also combine our analysis with more methods and schemes in optimization, e.g., adaptive gradient methods, gradient compression, and distributed optimization (Kingma & Ba, 2015; Agarwal et al., 2018; Zhou et al., 2020; Li et al., 2022).

ETHIC STATEMENT

This paper studied a class of private-preserving machine learning algorithms called differential private SGD. We provide improved convergence analysis and some new theoretical insights into DP-SGD. We are not aware of any potential negative social impact of this work. The datasets used for the experiments do not contain personally identifiable information or offensive content.

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

## 8 APPENDIX

### APPENDIX A LEMMAS

#### A.1 STANDARD FACTS

**Lemma A.1.** *For any $a, b \in \mathbb{R}$, it is true that $(a + b)^2 \leq 2a^2 + 2b^2$.*

**Lemma A.2.** *It holds that $\sum_{i=1}^{\infty}(i/2^i) = 2$.*

**Lemma A.3.** *Given $n$ random events $A_1, A_2, \ldots, A_n$. It holds that*

$$\Pr[\cup_{i=1}^n A_i] = \Pr[A_1] + \Pr[A_1^c \cap A_2] + \ldots + \Pr[\cap_{i=1}^{n-1} A_i^c \cap A_n].$$

#### A.2 SUB-GAUSSIAN AND SUB-EXPONENTIAL PROPERTIES

**Lemma A.4** (Sub-Gaussian properties, Vershynin, 2018, Proposition 2.5.2)**.** *Let $X$ be a random variable. Then the following properties are equivalent; the parameters $K_i, i = 1, \ldots, 4$ differ from each other by at most an absolute constant factor.*

- *The tails of $X$ satisfy*

$$\Pr[|X| \geq t] \leq 2\exp(-t^2/K_1^2) \qquad \forall t \geq 0.$$

- *The moment generating function (MGF) of $X^2$ satisfies*

$$\mathbb{E}[\exp(\lambda X^2)] \leq \exp(K_2^2 \lambda) \qquad \forall \lambda \in [0, 1/K_2^2].$$

- *The MGF of $X^2$ is bounded at some point, e.g.,*

$$\mathbb{E}[\exp(X^2/K_3^2)] \leq e.$$

- *$\mathbb{E}[X] = 0$ and the MGF of $X$ satisfies*

$$\mathbb{E}[\exp(\lambda X)] \leq \exp(K_4^2 \lambda^2) \qquad \forall \lambda \in \mathbb{R}.$$

**Lemma A.5** (Harvey et al., 2019, Lemma A.4)**.** *Let $X_1, X_2, \ldots, X_n$ be random variables. Assume that there exist $K_1, K_2, \ldots, K_n > 0$ such that $\mathbb{E}[\exp(\lambda X_i)] \leq \exp(\lambda K_i)$ for all $0 \leq \lambda \leq 1/K_i$. Then $\mathbb{E}[\exp(\lambda \sum_{i=1}^n X_i)] \leq \exp(\lambda \sum_{i=1}^n K_i)$ for all $0 \leq \lambda \leq 1/\sum_{i=1}^n K_i$.*

**Lemma A.6** (Harvey et al., 2019, Claim A.7)**.** *Suppose $X$ is a random variable such that there exists constants $c$ and $K$ such that $\mathbb{E}[\exp(\lambda X)] \leq c\exp(\lambda K)$ for all $0 \leq \lambda \leq 1/K$. Then for any $\delta \in (0, 1)$, $\Pr[X \geq K\log(1/\delta)] \leq ce\delta$.*

Note that the original lemmas from Harvey et al. (2019) assume $\mathbb{E}[\exp(\lambda X)] \leq \exp(\lambda K)$ for all $\lambda \leq 1/K$. It is easy to check that their results also hold for $X$ such that $\mathbb{E}[\exp(\lambda X)] \leq \exp(\lambda K)$ for all $0 \leq \lambda \leq 1/K$.

**Lemma A.7.** *Let $X_1, X_2, \ldots, X_T$ be random variables such that $\mathbb{E}[\exp(\lambda X_i)] \leq \exp(\lambda K) \, \forall \lambda \in [0, K]$ for all $i \in \{1, 2, \ldots, T\}$. Then there exists absolute positive constant $c$ such that $\Pr\left[\sum_{i=1}^T X_i \geq TK\log(1/\delta)\right] \leq ce\delta$ for any $\delta \in (0, 1)$.*

*Proof.* By Lemma A.5,

$$\mathbb{E}\left[\exp\left(\lambda \sum_{i=1}^T X_i\right)\right] \leq \exp(TK\lambda) \qquad \forall \lambda \in [0, (TK)^{-1}].$$

Then Lemma A.6 gives

$$\Pr\left[\sum_{i=1}^T X_i \geq TK\log(1/\delta)\right] \leq ce\delta \qquad \forall \delta \in (0, 1).$$

The above finishes the proof. $\qquad \square$

**Lemma A.8** (Tail bound for the maximum of sub-Gaussian variables). *Let $X_1, X_2, \ldots, X_n$ be random variables. Assume that there exist $K > 0$ such that $\mathbb{E}[\exp(\lambda X_i)] \leq \exp(\lambda^2 K^2)$ for all $\lambda \in \mathbb{R}$ and $i \in [n]$. Then there exist some absolute positive constant $c$ such that*

$$\Pr\left[\max_{i \in [n]} |X_i| \geq cK\sqrt{\log(2n)} + ct\right] \leq \exp(-t^2/K^2) \qquad \forall t \geq 0.$$

*Proof.* By Lemma A.4, there exist some absolute constant $c > 0$ such that

$$\Pr[|X_i| \geq t] \leq 2\exp\left(-t^2/(c^2 K^2)\right) \qquad \forall t \geq 0, i \in [n].$$

Then we simply apply union bound to obtain the conclusion. For any $t \geq 0$,

$$\Pr\left[\max_{i \in [n]} |X_i| \geq cK\sqrt{\log(2n)} + ct\right]$$

$$\leq \sum_{i=1}^{n} \Pr\left[|X_i| \geq cK\sqrt{\log(2n)} + ct\right]$$

$$\leq 2n \exp\left(-(cK\sqrt{\log(2n)} + ct)^2/(c^2 K^2)\right)$$

$$\leq \exp(\log(2n))\exp\left(-\frac{K^2\log(2n)}{K^2} - \frac{2tK\sqrt{\log(2n)}}{K^2} - \frac{t^2}{K^2}\right)$$

$$= \exp\left(-\frac{2tK\sqrt{\log(2n)}}{K^2} - \frac{t^2}{K^2}\right)$$

$$\leq \exp(-t^2/K^2),$$

which yields the desired result. $\qquad\qquad\square$

**Lemma A.9.** *Let $X$ be a random variable and $\mathcal{E}$ be a random event such that $\mathbb{E}[|X| \mid \mathcal{E}] \leq \alpha$ and $\Pr[\mathcal{E}] \geq 1 - \delta$ for some $\alpha > 0, \delta \in (0, 1)$. If $X$ satisfies $\Pr[|X| - \beta \geq t] \leq 2\exp(-t/K) \, \forall t \geq 0$ for some $\beta, K > 0$. Then*

$$\mathbb{E}[|X|] \leq \alpha + \delta\beta + \delta\log(8/\delta)K.$$

*Proof.* Let $Q : [0, 1] \to \mathbb{R} \cup \{\infty\}$ be the quantile function for the random variable $|X|$, e.g.,

$$Q(p) = \inf\left\{x \in \mathbb{R} \mid p \leq \Pr[|X| \leq x]\right\} \qquad \forall p \in [0, 1].$$

By the assumption that $|X| - \beta$ is sub-exponential, we have that

$$Q(1 - \delta') \leq \beta + K\log(2/\delta') \qquad \forall \delta' \in (0, 1). \tag{3}$$

Now we are ready to prove the conclusion. First notice that

$$\mathbb{E}[|X|] = \Pr[\mathcal{E}]\mathbb{E}[|X| \mid \mathcal{E}] + \Pr[\mathcal{E}^c]\mathbb{E}[|X| \mid \mathcal{E}^c]$$
$$\leq \alpha + \mathbb{E}[|X| \cdot \mathbf{1}_{\mathcal{E}^c}], \tag{4}$$

where $\mathbf{1}_{\mathcal{E}^c}$ is the indicator function with the event $\mathcal{E}^c$. The remaining is to bound $\mathbb{E}[|X| \cdot \mathbf{1}_{\mathcal{E}^c}]$. Define a new set of events $A_i := \left\{|X| \in \left[Q\left(1 - \frac{\delta}{2^{i-1}}\right), Q\left(1 - \frac{\delta}{2^i}\right)\right]\right\}, i = 1, 2, \ldots$, and denote the probability measure space as $(\Omega, \mathcal{F}, \mu)$. Then

$$\mathbb{E}[|X| \cdot \mathbf{1}_{\mathcal{E}^c}] = \int_\Omega |X| \cdot \mathbf{1}_{\mathcal{E}^c} d\mu(\omega)$$

$$\leq \int_\Omega |X| \cdot \mathbf{1}_{|X| \geq Q(1-\delta)} d\mu(\omega)$$

$$\leq \sum_{i=1}^\infty \int_\Omega |X| \cdot \mathbf{1}_{A_i} d\mu(\omega)$$

$$\leq \sum_{i=1}^\infty \frac{\delta}{2^i} Q\left(1 - \frac{\delta}{2^i}\right)$$

$$\overset{(i)}{\leq} \; \delta\beta + \sum_{i=1}^{\infty} \frac{\delta}{2^i} K \log\left(\frac{2^{i+1}}{\delta}\right)$$

$$\leq \; \delta\beta + \delta K\left(\sum_{i=1}^{\infty} \frac{1}{2^i}((i+1)\log(2) + \log(1/\delta))\right)$$

$$\overset{(ii)}{\leq} \; \delta\beta + \delta K(3\log(2) + \log(1/\delta)),$$

where (i) is by eq. (3) and (ii) is by the standard scalar inequality (Lemma A.2). The above together with eq. (4) yields the desired result. $\square$

### A.3 THE GENERALIZED FREEDMAN INEQUALITY

We restate the generalized Freedman inequality and its corollary from Harvey et al. (2019).

**Lemma A.10** (Generalized Freedman Inequality, Harvey et al., 2019, Theorem 3.2). *Let $\{d_i, \mathcal{F}_i\}_{i=1}^{T}$ be a martingale difference sequence. Suppose $v_{i-1} \geq 0, \forall i \in [T]$ are $\mathcal{F}_{i-1}$-measurable random variables such that $\mathbb{E}[\exp(\lambda d_i) \mid \mathcal{F}_{i-1}] \leq \exp(\frac{\lambda^2}{2}v_{i-1})$ for all $i \in [T], \lambda > 0$. Let $S_t = \sum_{i=1}^{t} d_i$ and $V_t = \sum_{i=1}^{t} v_{i-1}$. Let $\alpha_i \geq 0$ and $\alpha = \max_{i \in [T]} \alpha_i$. Then*

$$\Pr\left[\bigcup_{t=1}^{T}\left\{S_t \geq x \quad and \quad V_t \leq \sum_{i=1}^{t} \alpha_i d_i + \beta\right\}\right] \; \leq \; \exp\left(-\frac{x}{4\alpha + 8\beta/x}\right) \quad \forall x, \beta > 0.$$

The following lemma is an immediate consequence from the generalized Freedman inequality.

**Lemma A.11.** *Let $\{d_i, \mathcal{F}_i\}_{i=1}^{T}$ be a martingale difference sequence. Suppose $v_{i-1} \geq 0, \forall i \in [T]$ are $\mathcal{F}_{i-1}$-measurable random variables such that $\mathbb{E}[\exp(\lambda d_i) \mid \mathcal{F}_{i-1}] \leq \exp(\frac{\lambda^2}{2}v_{i-1})$ for all $i \in [T], \lambda > 0$. Let $S_t = \sum_{i=1}^{t} d_i$ and $V_t = \sum_{i=1}^{t} v_{i-1}$. Let $\delta \in (0,1)$ and suppose there are positive values $R(\delta) > 0$ and non-negative values $\{\alpha_i^{(t)}, i = 1, 2, \ldots, T, t = 1, 2, \ldots, T\}$ such that*

$$\Pr\left[\bigcap_{t=1}^{T}\left\{V_t \leq \sum_{i=1}^{t} \alpha_i^{(t)} d_i + R(\delta)\right\}\right] \; \geq \; 1 - \delta.$$

*Let $\alpha = \max_{i \in [T], t \in [T]} \alpha_i^{(t)}$. Then*

$$\Pr\left[\bigcup_{t=1}^{T}\{S_t \geq x\}\right] \; \leq \; \delta + T\exp\left(-\frac{x}{4\alpha + 8R(\delta)/x}\right) \quad \forall x, \beta > 0.$$

*Proof.* Given $\delta \in (0,1), x \in \mathbb{R}$. Define the events $A_t := \{S_t \geq x\}$ and $B_t := \left\{V_t \leq \sum_{i=1}^{t} \alpha_i^{(t)} d_i + R(\delta)\right\}$. Then

$$\Pr\left[\bigcup_{t=1}^{T}\{S_t \geq x\}\right] \; = \; \Pr\left[\bigcup_{t=1}^{T} A_t\right]$$

$$= \; \Pr\left[\left(\bigcup_{t=1}^{T} A_t\right)\bigcap\left(\bigcap_{t=1}^{T} B_t\right)\right] + \Pr\left[\left(\bigcup_{t=1}^{T} A_t\right)\bigcap\left(\bigcap_{t=1}^{T} B_t\right)^c\right]$$

$$\overset{(i)}{\leq} \; \Pr\left[\left(\bigcup_{t=1}^{T} A_t\right)\bigcap\left(\bigcap_{t=1}^{T} B_t\right)\right] + \delta$$

$$\leq \; \Pr\left[\bigcup_{t=1}^{T}\left(A_t\bigcap\left(\bigcap_{i=1}^{T} B_i\right)\right)\right] + \delta$$

$$\leq \; \Pr\left[\bigcup_{t=1}^{T}\left(A_t\bigcap B_t\right)\right] + \delta$$

---

**Algorithm 4** Stochastic gradient descent

---
1: **Input:** total iterations $T \in \mathbb{N}$, learning rate $\eta > 0$, initial iterate $w^{(0)}$.
2: **for** $t \leftarrow 0, \ldots, T-1$ **do**
3:      $w^{(t+1)} = w^{(t)} - \eta \left( \nabla f(w^{(t)}) + \xi^{(t)} \right)$;
4: **end for**
5: **Return:** $w^{(i)}$ where $i$ uniform randomly sampled from $\{0, 1 \ldots, T\}$.

---

$$\overset{(ii)}{\leq} \sum_{t=1}^{T} \Pr \left[ A_t \bigcap B_t \right] + \delta$$

$$\overset{(iii)}{\leq} \delta + T \exp \left( - \frac{x}{4\alpha + 8R(\delta)/x} \right) \qquad \forall x, \beta > 0,$$

where (i) is by the assumption $\Pr \left[ \cap_{t=1}^{T} B_t \right] \geq 1 - \delta$, (ii) is by union bound (note that we can not directly use Lemma A.10 since $V_t$ relies on $\alpha_1^{(t)}, \ldots, \alpha_t^{(t)}$ instead of $\alpha_1, \ldots, \alpha_t$), (iii) is by applying Lemma A.10 for $T$ times. $\qquad \square$

## A.4 OTHER LEMMAS

Recall Lemma 3.2.

*Proof of Lemma 3.2.* By the smoothness of $h$, we have

$$h(v) \leq h(u) + \langle \nabla h(u), v - u \rangle + \frac{L}{2} \|v - u\|^2 \qquad u, v \in \mathbb{R}^d.$$

Making the identification that $v = u - \nabla h(u)/L$. We obtain

$$h(u - \nabla h(u)/L) \leq h(u) - \frac{1}{2L} \|\nabla h(u)\|^2$$

$$\implies \inf_{v \in \mathbb{R}^d} h(v) \leq h(u - \nabla h(u)/L) \leq h(u) - \frac{1}{2L} \|\nabla h(u)\|^2$$

$$\implies \|\nabla h(u)\|^2 \leq 2L \left( h(u) - \inf_{v \in \mathbb{R}^d} h(v) \right) \qquad \forall u \in \mathbb{R}^d.$$

$\qquad \square$

Next, we review the standard convergence rate of SGD algorithm in the following lemma.

**Lemma A.12** (Ghadimi & Lan, 2013, Theorem 2.1). *Denote $\hat{w}$ as the output of Algorithm 4. Assume that $f_i$'s are L-smooth for some $L > 0$, $\eta \leq 1/L$ and there exist $\sigma \geq 0$ such that $\mathbb{E}[\|\xi^{(t)}\|^2] \leq \sigma^2$ for all $t \in \mathbb{N}$.*

- *If $f_i$'s are further convex, then*

$$\mathbb{E}[f(\hat{w}) - f^*] \leq \frac{\inf_{w \in \mathcal{W}^*} \|w^{(0)} - w\|^2}{(T+1)\eta} + \sigma^2 \eta,$$

  *where $\mathcal{W}^*$ is the set of solutions.*

- *If $f_i$'s are not necessarily convex, then*

$$\mathbb{E} \left[ \|\nabla f(\hat{w})\|^2 \right] \leq \frac{2 \left( f(w^{(0)}) - f^* \right)}{(T+1)\eta} + L\sigma^2 \eta.$$

The following lemma connects the WGC parameters of $f_i$'s to the WGC parameters of $f$.

**Lemma A.13.** *Assume that $f_i$'s are $(\beta_1, \beta_2)$-WGC for all $i \in [n]$. Then $f := \frac{1}{n} \sum_{i=1}^{n} f_i$ is $(\beta_1, \beta_2 + \Gamma\beta_1)$-WGC, where $\Gamma = \frac{1}{n} \sum_{i=1}^{n} (f^* - f_i^*)$.*

*Proof.*

$$\|\nabla f(w)\|^2 \overset{\text{(i)}}{\leq} \frac{1}{n} \sum_{i=1}^n \|\nabla f_i(w)\|^2$$

$$\overset{\text{(ii)}}{\leq} \frac{1}{n} \sum_{i=1}^n \beta_1 (f_i(w) - f_i^*) + \beta_2$$

$$\leq \beta_1 (f(w) - f^*) + \Gamma \beta_1 + \beta_2,$$

where (i) is by the convexity of $\|\cdot\|^2$ and (ii) is by the assumption that $f_i$'s are $(\beta_1, \beta_2)$-WGC. $\qquad\square$

## APPENDIX B    PROOFS FOR SECTION 4

### B.1    PROOF OF PROPOSITION 4.2

*Proof.* We begin with the smoothness of $f$,

$$f(w^{(t+1)}) \leq f(w^{(t)}) + \langle \nabla f(w^{(t)}), w^{(t+1)} - w^{(t)} \rangle + \frac{L}{2} \|w^{(t+1)} - w^{(t)}\|^2$$

$$= f(w^{(t)}) - \eta \|\nabla f(w^{(t)})\|^2 - \eta \langle \nabla f(w^{(t)}), \zeta^{(t)} \rangle + \frac{L\eta^2}{2} \|\nabla f(w^{(t)}) + \zeta^{(t)}\|^2$$

$$= f(w^{(t)}) - \underbrace{\left( \eta - \frac{L\eta^2}{2} \right)}_{\geq 0} \|\nabla f(w^{(t)})\|^2 + \underbrace{\frac{L\eta^2}{2} \|\zeta^{(t)}\|^2}_{z_t} + \underbrace{(L\eta^2 - \eta)\langle \nabla f(w^{(t)}), \zeta^{(t)} \rangle}_{u_t}.$$

$$(5)$$

Let

$$Z_t := \sum_{i=0}^t z_i, \qquad U_t := \sum_{i=0}^t u_i.$$

Recursively apply eq. (5) gives

$$f(w^{(t)}) \leq f(w^{(0)}) + Z_{T-1} + U_{t-1}, \qquad \forall t = 0, 1, \ldots, T. \qquad (6)$$

All we need is to show that $Z_{T-1}$ and $U_{t-1}$ are bounded above with high probability. First, we bound $Z_{T-1}$. Notice that $\zeta^{(t)}$'s are sub-Gaussian variables, therefore $Z_{T-1}$ is sub-exponential. We apply Lemma A.7, there exist some absolute constant $c_1 > 0$ such that for any $\delta' \in (0, 0.5)$,

$$Z_{T-1} \leq \frac{L\eta^2}{2} \left( c_1 T \tilde{\sigma}^2 \log(1/\delta') \right)$$

$$\leq \frac{c_1 L \log(1/\delta')}{2} \qquad \text{(By the definition of } \eta) \qquad (7)$$

with probability at least $1 - \delta'$.

Next we bound the term $U_{t-1}$. Noticing that $\{U_t\}_{t=0}^\infty$ is a martingale sequence, we bound it by the generalized Freedman inequality (Lemma A.10). Denote $\mathcal{F}_{t-1}$ to be the $\sigma$-Algebra generated from $\{w^{(1)}, \ldots, w^{(t)}\}$. Then by the definition of $\zeta^{(t)}$, we have

$$\mathbb{E}\left[ \exp \left( u_t^2 / ((L\eta^2 - \eta)^2 \tilde{\sigma}^2 \|\nabla f(w^{(t)})\|^2) \right) \mid \mathcal{F}_{t-1} \right] \leq e,$$

where we use the inequality $\|u_t\|^2 \leq (L\eta^2 - \eta)^2 \|\nabla f(w^{(t)})\|^2 \|\zeta^{(t)}\|^2$. By the properties of sub-Gaussian variable (Lemma A.4), there exist some absolute constant $c_2 > 0$ such that

$$\mathbb{E}\left[ \exp \left( \lambda u_t \right) \mid \mathcal{F}_{t-1} \right] \leq \exp \left( \frac{\lambda^2}{2} c_2 (L\eta^2 - \eta)^2 \tilde{\sigma}^2 \|\nabla f(w^{(t)})\|^2 \right) \quad \forall \lambda \in \mathbb{R}.$$

Let $v_{t-1} := c_2 (L\eta^2 - \eta)^2 \tilde{\sigma}^2 \|\nabla f(w^{(t)})\|^2$. Then

$$v_{t-1} \leq 2 c_2 L \eta^2 \tilde{\sigma}^2 (f(w^{(t)}) - f^*) \qquad \text{(By the weak growth condition)}$$

$$\leq \frac{2 c_2 L}{T} (f(w^{(t)}) - f^*) \qquad \text{(By } \eta \leq 1/(\tilde{\sigma}\sqrt{T})) \quad .$$

Furthermore, for any $t = 0, 1, \ldots, T-1$,

$$
\begin{aligned}
V_t &:= \sum_{i=0}^{t} v_{i-1} \\
&\leq c_2 \sum_{i=0}^{t} \left( \frac{2L}{T} (f(w^{(i)}) - f^*) \right) \\
&\leq c_2 \sum_{i=0}^{t} \frac{2L}{T} \left( f(w^{(0)}) - f^* + Z_{T-1} + U_{i-1} \right) \qquad \text{(By eq. (6))} \\
&\leq 2c_2 L(f(w^{(0)}) - f^* + Z_{T-1}) + \frac{2c_2 L}{T} \sum_{i=0}^{t} \sum_{j=0}^{i-1} u_j \\
&\leq 2c_2 L(f(w^{(0)}) - f^* + Z_{T-1}) + \frac{2c_2 L}{T} \sum_{i=0}^{t} (t-i) u_i \qquad \text{(Rearranging)} \\
&\leq 2c_2 L(f(w^{(0)}) - f^* + Z_{T-1}) + 2c_2 L \sum_{i=0}^{t} \frac{t-i}{T} u_i.
\end{aligned}
$$

Combining the above with eq. (7), with probability $1 - \delta'$

$$
V_t \leq 2c_2 L \left( f(w^{(0)}) - f^* + c_1 L \log(1/\delta')/2 \right) + 2c_2 L \sum_{i=0}^{t} \frac{t-i}{T} u_i \qquad \forall t = 0, 1, \ldots, T-1.
$$

Now we are ready to apply Lemma A.11. Making the identification

$$
d_i = u_i, \; \alpha_i^{(t)} = 2c_2 L \frac{t-i}{T}, \; \alpha = 2c_2 L, \; R(\delta') = 2c_2 L \left( f(w^{(0)}) - f^* + c_1 L \log(1/\delta')/2 \right)
$$

and apply Lemma A.11 gives

$$
\Pr \left[ \bigcup_{t=0}^{T-1} \{ U_t \geq x \} \right] \leq \delta' + T \exp \left( -\frac{x}{4\alpha + 8R(\delta')/x} \right) \qquad \forall x > 0.
$$

It is easy to verify that with the choice $x = \max \left\{ 4\sqrt{R(\delta') \log(T/\delta')}, 8\alpha \log(T/\delta') \right\}$, we have

$$
\Pr \left[ \bigcup_{t=0}^{T-1} \{ U_t \geq x \} \right] \leq 2\delta'. \tag{8}
$$

Therefore, with probability at least $1 - 2\delta'$,

$$
\begin{aligned}
\max_{t=0,1,\ldots,T} &f(w^{(t)}) - f^* \\
&\overset{(i)}{\leq} f(w^{(0)}) - f^* + c_1 L \log(1/\delta')/2 \\
&\quad + \max \left\{ 4\sqrt{(2c_2 L(f(w^{(0)}) - f^* + c_1 L \log(1/\delta')/2)) \log(T/\delta')}, 16 c_2 L \log(T/\delta') \right\} \\
&\overset{(ii)}{\leq} \left( \sqrt{f(w^{(0)}) - f^* + c_1 L \log(1/\delta')/2} + 4\sqrt{c_2 L \log(T/\delta')} \right)^2 \\
&\overset{(iii)}{\leq} 2 \left( f(w^{(0)}) - f^* + c_1 L \log(1/\delta')/2 \right) + 32 c_2 L \log(T/\delta'),
\end{aligned}
$$

where (i) is by eq. (6) and eq. (8), (ii) is by the fact that $\max\{a, b\} \leq a + b$ and $\sqrt{a+b} \leq \sqrt{a} + \sqrt{b}$ for any $a, b \geq 0$, (iii) is by Lemma A.1. Substitute $\delta'$ with $\delta/2$, we obtain the desired result. $\qquad\square$

## B.2 PROOF OF PROPOSITION 4.3

*Proof.* By the assumption that $f_i$'s are $L$-smooth, we know that $f$ is also $L$-smooth. Therefore $f$ is $(2L, 0)$-WGC (By Lemma 3.2), e.g.,

$$\|\nabla f(w^{(t)})\| \leq \sqrt{2L\left(f(w^{(t)}) - f^*\right)} \qquad \forall t = 0, 1, \ldots, T.$$

Apply Proposition 4.2. We obtain that

$$\max_{t=0,1,\ldots,T} \|\nabla f(w^{(t)})\| \leq \sqrt{4L\left(f(w^{(0)}) - f^* + c_1 L \log(2/\delta)/2\right) + 64 c_2 L^2 \log(2T/\delta)} \quad (9)$$

with probability at least $1 - \delta$ for some absolute positive constants $c_1, c_2$.

The remaining is to bound the individual gradient norm $\|\nabla f_i(w^{(t)})\|$. When Assumption 4.1 holds, we have that

$$\max_{i \in [n], t=0,1,\ldots,T} \|\nabla f_i(w^{(t)})\| \leq \max_{t \in \{0,1\ldots,T\}} \|\nabla f(w^{(t)})\| + \max_{i \in [n], t=0,1,\ldots,T} \|\nabla f_i(w^{(t)}) - f(w^{(t)})\|.$$

By Assumption 4.1, we know that $\|\nabla f_i(w) - \nabla f(w)\|$ is sub-Gaussian for all $w \in \mathbb{R}^d$. Therefore we can apply Lemma A.8 to bound the maximum of sub-Gaussian variables, which gives

$$\max_{i \in [n], t=0,1,\ldots,T} \|\nabla f_i(w^{(t)}) - f(w^{(t)})\| < c_3 \rho \sqrt{\log(n(T+1))} + c_3 \sqrt{\rho \log(1/\delta)} \quad (10)$$

for some absolute positive constant $c_3 > 0$ and any $\delta \in (0, 1)$.

Combining eq. (9) and eq. (10). We obtain that

$$\max_{i \in [n], t=0,1,\ldots,T} \|\nabla f_i(w^{(t)})\| \leq \sqrt{4L\left(f(w^{(0)}) - f^* + c_1 L \log(2/\delta)/2\right) + 64 c_2 L^2 \log(2T/\delta)}$$
$$+ c_3 \rho \sqrt{\log(n(T+1))} + c_3 \sqrt{\rho \log(1/\delta)}$$

with probability at least $1 - 2\delta$, which yields the desired result eq. (1) under Assumption 4.1.

$\square$

## B.3 PROOF OF PROPOSITION 4.4

*Proof.* We first consider the case that $f_i$'s are convex and smooth. By Lemma 3.2, we know that $f_i$'s are $(2L, 0)$-WGC in this scenario. Setting

$$C = \sqrt{4L\left(f(w^{(0)}) - f^* + c_1 L \log(2/\delta')/2 + 16 c_2 L \log(2T/\delta')\right)}$$
$$+ c_3 \rho \sqrt{\log(n(T+1))} + c_3 \sqrt{\rho \log(1/\delta')},$$
$$\sigma = c_4 \frac{q\sqrt{T \log(1/\delta)}}{\epsilon}, \quad (11)$$

where $c_1, c_2, c_3$ correspond to the absolute positive constants that appeared in the proof of Proposition 4.2, $c_4$ is some positive constant that appeared in Theorem 3.4.

Next we need to apply Proposition 4.3. Denote $\mathcal{B}_t$ as the batch sampled at the $t$-th iteration and let

$$\zeta^{(t)} := \frac{1}{|\mathcal{B}_t|}\left(\sum_{i \in \mathcal{B}_t} \nabla f_i(w^{(t)}) + \xi^{(t)}\right) - \nabla f(w^{(t)}).$$

By the definition of $\xi^{(t)}$ and Assumption 4.1, $\zeta^{(t)}$'s satisfy

$$\mathbb{E}\left[\|\zeta^{(t)}\|^2 / \left(\frac{c_5 \rho^2}{B} + \frac{c_5 C^2 d\sigma^2}{B^2}\right)\right] \leq e \qquad \text{(by the property of Poisson sampling)} \quad,$$

where $c_5$ is some positive absolute constant. Let

$$\tilde{\sigma}^2 = c_5 \left(\frac{\rho^2}{B} + \frac{C^2 d\sigma^2}{B^2}\right), \quad \eta = \min\left\{\frac{1}{2L}, \frac{1}{\tilde{\sigma}\sqrt{T}}\right\}. \quad (12)$$

Then Proposition 4.3 tells us that Algorithm 2 with the above setup of $\tilde{\sigma}$ and $\eta$ will produce iterates such that

$$\|\nabla f_i(w^{(t)})\| \leq C, \qquad \forall i \in [n], t = 0, 1, \ldots, T,$$

with probability $1 - \delta'$.

The above analysis is based on Algorithm 2. Next, we draw the connection between Algorithm 2 with the above setup of parameters and Algorithm 1 with the parameter setup in eq. (11) and $\eta$ defined as in eq. (12). We introduce some new notation to help our analysis. To distinguish between Algorithm 2 (SGD) and Algorithm 1 (DP-SGD-GC), we denote $\{\tilde{w}^{(t)}\}_{t=0}^T$ as the iterates from Algorithm 2 and $\{w^{(t)}\}_{t=0}^T$ as the iterates from Algorithm 1. We further let $\widetilde{\mathcal{E}}_t$ and $\mathcal{E}_t$ as the event $\{\|\nabla f_i(\tilde{w}^{(t)})\| \leq C, \forall i \in [n]\}$ and $\{\|\nabla f_i(w^{(t)})\| \leq C, \forall i \in [n]\}$ respectively. For two random variables $A$ and $B$, we denote $A \sim B$ if $A$ and $B$ are independent and identically distributed. Consider two independent runs of Algorithm 2 and Algorithm 1 with $\tilde{w}^{(0)} = w^{(0)}$, we are going to show that

- $[\tilde{w}^{(0)}, \ldots, \tilde{w}^{(T)}]$ conditioned on the event $\cap_{t=0}^T \widetilde{\mathcal{E}}_t$ has the same distribution as $[w^{(0)}, \ldots, w^{(T)}]$ conditional on the event $\cap_{t=0}^T \mathcal{E}_t$;

- $\Pr[\cap_{t=0}^T \widetilde{\mathcal{E}}_t] = \Pr[\cap_{t=0}^T \mathcal{E}_t]$.

The first conclusion should be obvious. Given $\tilde{w}^{(0)} = w^{(0)}$, we can conclude that $\tilde{w}^{(1)} \sim w^{(1)}$ conditioned on $\widetilde{\mathcal{E}}_0$ and $\mathcal{E}_0$ (both gradients are bounded), which further implies $\tilde{w}^{(2)} \sim w^{(2)}$ conditioned on $\widetilde{\mathcal{E}}_1$ and $\mathcal{E}_1$ and so on.

For the second conclusion, we prove by induction. Given $\tilde{w}^{(0)} = w^{(0)}$, the base case $\Pr[\widetilde{\mathcal{E}}_0] = \Pr[\mathcal{E}_0]$ is obviously true. Next, given $\Pr[\cap_{t=0}^m \widetilde{\mathcal{E}}_t] = \Pr[\cap_{t=0}^m \mathcal{E}_t]$, we are going to prove $\Pr[\cap_{t=0}^{m+1} \widetilde{\mathcal{E}}_t] = \Pr[\cap_{t=0}^{m+1} \mathcal{E}_t]$. Knowing that

$$\Pr[\cap_{t=0}^{m+1} \widetilde{\mathcal{E}}_t] = 1 - \Pr[\cup_{t=0}^{m+1} \widetilde{\mathcal{E}}_t^c]$$
$$= 1 - \left( \Pr[\widetilde{\mathcal{E}}_0^c] + \Pr[\widetilde{\mathcal{E}}_0 \cap \widetilde{\mathcal{E}}_1^c] + \ldots + \Pr[\cap_{t=0}^m \widetilde{\mathcal{E}}_t \cap \widetilde{\mathcal{E}}_{m+1}^c] \right) \qquad \text{(by Lemma A.3)} \quad .$$

We only need to prove that $\Pr[\cap_{t=0}^p \widetilde{\mathcal{E}}_t \cap \widetilde{\mathcal{E}}_{p+1}^c] = \Pr[\cap_{t=0}^p \mathcal{E}_t \cap \mathcal{E}_{p+1}^c]$ $\forall p = 0, 1, \ldots, m$, which is equivalent to $\Pr[\widetilde{\mathcal{E}}_{p+1}^c \mid \cap_{t=0}^p \widetilde{\mathcal{E}}_t] \Pr[\cap_{t=0}^p \widetilde{\mathcal{E}}_t] = \Pr[\mathcal{E}_{p+1}^c \mid \cap_{t=0}^p \mathcal{E}_t] \Pr[\cap_{t=0}^p \mathcal{E}_t]$. By induction, suppose $\Pr[\cap_{t=0}^p \widetilde{\mathcal{E}}_t] = \Pr[\cap_{t=0}^p \mathcal{E}_t]$ holds. Conditioning on the two events $\cap_{t=0}^p \widetilde{\mathcal{E}}_t$ and $\cap_{t=0}^p \mathcal{E}_t$, it is obvious that $\tilde{w}^{(p+1)} \sim w^{(p+1)}$ and therefore $\Pr[\widetilde{\mathcal{E}}_{p+1}^c \mid \cap_{t=0}^p \widetilde{\mathcal{E}}_t] = \Pr[\mathcal{E}_{p+1}^c \mid \cap_{t=0}^p \mathcal{E}_t]$. Combining the above together, we prove that $\Pr[\cap_{t=0}^{m+1} \mathcal{E}_t]$ and finish the induction.

With the above two conclusions, we can transfer the convergence of Algorithm 2 conditioned on the event $\cap_{t=0}^T \widetilde{\mathcal{E}}_t$ to Algorithm 1 conditioned on the event $\cap_{t=0}^T \mathcal{E}_t$.

Next, the conditional convergence of SGD. The proof is simply based on the law of total expectation. Denote $\tilde{w}_{\text{out}}$ as the output of Algorithm 2 and $\widetilde{\mathcal{E}} = \cap_{t=0}^T \widetilde{\mathcal{E}}_t$.

$$\mathbb{E}[f(\tilde{w}_{\text{out}}) - f^* \mid \widetilde{\mathcal{E}}] \Pr[\widetilde{\mathcal{E}}] + \mathbb{E}[f(\tilde{w}_{\text{out}}) - f^* \mid \widetilde{\mathcal{E}}^c] \Pr[\widetilde{\mathcal{E}}^c] = \mathbb{E}[f(\tilde{w}_{\text{out}}) - f^*]$$
$$\implies \mathbb{E}[f(\tilde{w}_{\text{out}}) - f^* \mid \widetilde{\mathcal{E}}] \Pr[\widetilde{\mathcal{E}}] \leq \mathbb{E}[f(\tilde{w}_{\text{out}}) - f^*]$$
$$\overset{(i)}{\implies} \mathbb{E}[f(\tilde{w}_{\text{out}}) - f^* \mid \widetilde{\mathcal{E}}] \leq 2\mathbb{E}[f(\tilde{w}_{\text{out}}) - f^*], \tag{13}$$

where (i) is by the fact that $\Pr[\widetilde{\mathcal{E}}] \geq 1 - \delta' \geq 0.5$.

Now we can transfer the convergence of conditional SGD to conditional DP-SGD-GC. Notice that the term $\mathbb{E}[\|\xi^{(t)}\|^2]$ in Lemma A.12 is bounded by $\tilde{\sigma}^2$, then the classic convergence rate of SGD

(Lemma A.12) gives that

$$
\begin{aligned}
& \mathbb{E}\left[f(w_{\mathrm{priv}}) - f^* \mid \mathcal{E}\right] \\
=\ & \mathbb{E}\left[f(\tilde{w}_{\mathrm{out}}) - f^* \mid \widetilde{\mathcal{E}}\right] \\
\leq\ & 2\mathbb{E}\left[f(\tilde{w}_{\mathrm{out}}) - f^*\right] \qquad \text{(by eq. (13))} \\
\leq\ & \frac{2D_w^2}{(T+1)\eta} + 2\tilde{\sigma}^2\eta \qquad \text{(by Lemma A.12)} \\
\overset{(i)}{\leq}\ & \frac{4LD_w^2}{T} + \frac{2D_w^2\tilde{\sigma}}{\sqrt{T}} + \frac{2\tilde{\sigma}}{\sqrt{T}} \\
\overset{(ii)}{\leq}\ & \mathcal{O}\left( \frac{2LD_w^2}{T} + \frac{D_w^2\rho}{\sqrt{BT}} + \frac{D_w^2 C\sigma\sqrt{d}}{B\sqrt{T}} + \frac{\rho}{\sqrt{TB}} + \frac{C\sigma\sqrt{d}}{B\sqrt{T}} \right) \\
\overset{(iii)}{\leq}\ & \mathcal{O}\left( \frac{2LD_w^2}{T} + \frac{D_w^2\rho}{\sqrt{BT}} + \frac{D_w^2 Cq\sqrt{d\log(1/\delta)}}{B\epsilon} + \frac{\rho}{\sqrt{TB}} + \frac{Cq\sqrt{d\log(1/\delta)}}{B\epsilon} \right) \\
\overset{(iv)}{=}\ & \mathcal{O}\left( \frac{2LD_w^2}{T} + \frac{D_w^2\rho}{\sqrt{BT}} + \frac{D_w^2 C\sqrt{d\log(1/\delta)}}{n\epsilon} + \frac{\rho}{\sqrt{TB}} + \frac{C\sqrt{d\log(1/\delta)}}{n\epsilon} \right) \\
\overset{(v)}{=}\ & \mathcal{O}\left( \frac{D_w^2}{T} + \frac{D_w^2\rho}{\sqrt{BT}} + \frac{D_w^2(\log(1/\delta') + \log(n) + \log(T))\sqrt{d\log(1/\delta)}}{n\epsilon} \right),
\end{aligned}
$$

where (i) is by the definition of $\eta$, (ii) is by the definition of $\tilde{\sigma}$, (iii) is by the definition of $\sigma$ (eq. (11)), (iv) is by the definition of $q = B/n$, (v) is by noticing $C = \mathcal{O}(\log(1/\delta') + \log(n) + \log(T))$ from eq. (11).

When $f_i$'s are $L$-smooth but not necessarily convex. We set the parameters the same as for the convex case. Apply the convergence rate of SGD for smooth but not necessarily convex functions (Lemma A.12). Following the same proof template as for the convex case, can we obtain the desired result. $\qquad\square$

## B.4 Proof of Theorem 4.5

*Proof.* The outline of the proof is summarized as follows.

- First we show that the excess empirical risk e.g., $f(w_{\mathrm{priv}}) - f^*$ or gradient norm square e.g.,$\|\nabla f(w_{\mathrm{priv}})\|^2$ are sub-exponential random variables. We also derive a pessimistic upper bound of their sub-exponential parameters (Lemma B.1)[3].

- Next, we develop a technical tool to convert conditional expected error bound (the conditioning event happens with high probability) to expected error bound for sub-exponential random variables (Lemma A.9).

- Finally, by tuning $\delta'$ (the probability that gradient clipping happens during training; see Proposition 4.4), Lemma A.9 together with Lemma B.1 and Proposition 4.4 yield the desired result.

---

[3]We do not try to optimize the bound in Lemma B.1 as we show that a loose bound is enough for our purpose. The bound in Lemma B.1 should be improvable.

Now we proceed to the detailed proof. Consider the case that $f_i$'s are convex and $L$-smooth. Let $T, \sigma, C, \eta$ satisfy eq. (2). We apply Lemma A.9 by making the identification

$$
\begin{aligned}
X &:= f(w_{\mathrm{priv}}) - f^*; \\
\alpha &:= \mathcal{O}\left( \frac{D_w^2}{T} + \frac{D_w^2}{\sqrt{BT}} + \frac{D_w^2\left(\sqrt{\log(1/\delta')} + \sqrt{\log(Tn)}\right)\sqrt{d\log(1/\delta)}}{n\epsilon} \right) \\
\beta &:= 2(f(w^{(0)}) - f^*) + \frac{T^2 C^2}{L} \\
K &:= \frac{T^2 C^2 d\sigma^2}{c_1 L B^2},
\end{aligned}
$$

where the choice of $\alpha$ is by Proposition 4.4 and the choice of $\beta$ and $K$ is by Lemma B.1. Then Lemma A.9 gives that

$$
\begin{aligned}
\mathbb{E}[|X|] &= \alpha + \delta'\beta + \delta'\log(8/\delta')K \\
&\overset{(i)}{=} \widetilde{\mathcal{O}}\left( \frac{1}{T} + \frac{1}{\sqrt{BT}} + \frac{\sqrt{d}}{n\epsilon} \right),
\end{aligned}
$$

where (i) is true by setting

$$
\delta' = \min\left\{ \beta^{-1}\left( \frac{1}{T} + \frac{1}{\sqrt{BT}} + \frac{\sqrt{d}}{n\epsilon} \right), K^{-1}\left( \frac{1}{T} + \frac{1}{\sqrt{BT}} + \frac{\sqrt{d}}{n\epsilon} \right) \right\}.
$$

Note that the above proof template works as long as we can construct $\beta, K$ that polynomially depends on $T, n, d, \epsilon$ — we can tune $\delta'$ to convert all polynomial terms in $\beta$ and $K$ to polylogarithm terms by Lemma A.9. We find Lemma A.9 a quite convenient technical tool for our analysis and we are not aware if this result exist in the literature.

When $f_i$'s are smooth but not necessarily convex. The proof follows exactly the same as the convex case (just need to let $X := \|\nabla f(w_{\mathrm{priv}})\|^2, \beta := 4L(f(w^{(0)}) - f^*) + 2T^2 C^2, K = \frac{2T^2 C^2 d\sigma^2}{c_1 B^2}$ and apply eq. (15) instead of eq. (14)). We omit the details to avoid tedious repetition. □

**Lemma B.1** (A pessimistic bound for DP-SGD-GC)**.** *Assume that $f_i$'s are $L$-smooth. Let $w_{\mathrm{priv}}$ be the output from Algorithm 1 with $\eta \leq 1/L$. Then*

$$
\Pr\left[ f(w_{\mathrm{priv}}) - f^* \geq 2(f(w^{(0)}) - f^*) + \frac{T^2 C^2}{L} + t \right] \leq 2\exp\left( -\frac{tc_1 LB^2}{T^2 C^2 d\sigma^2} \right) \quad \forall t \geq 0. \quad (14)
$$

*Furthermore,*

$$
\Pr\left[ \|\nabla f(w_{\mathrm{priv}})\|^2 \geq 4L(f(w^{(0)}) - f^*) + 2T^2 C^2 + t \right] \leq 2\exp\left( -\frac{tc_1 B^2}{2T^2 C^2 d\sigma^2} \right) \quad \forall t \geq 0.
$$

$$(15)$$

*Proof.* We begin with the smoothness of $f$, for any $t \in \{0, 1, \ldots, T\}$

$$
\begin{aligned}
f(w^{(t)}) &\le f(w^{(0)}) + \langle \nabla f(w^{(0)}), w^{(t)} - w^{(0)} \rangle + \frac{L}{2} \|w^{(t)} - w^{(0)}\|^2 \\
&\overset{(i)}{\le} f(w^{(0)}) + \frac{1}{2L} \|\nabla f(w^{(0)})\|^2 + L\|w^{(t)} - w^{(0)}\|^2 \\
&\le f(w^{(0)}) + \frac{1}{2L} \|\nabla f(w^{(0)})\|^2 + L \left\| \eta \sum_{i=0}^{t-1} \left( \frac{1}{B} \sum_{j \in \mathcal{B}_t} \tilde{g}_j^{(i)} + \frac{1}{B} \xi^{(i)} \right) \right\|^2 \\
&\overset{(ii)}{\le} f(w^{(0)}) + \frac{1}{2L} \|\nabla f(w^{(0)})\|^2 + L\eta^2 T \sum_{i=0}^{T-1} \left\| \frac{1}{B} \sum_{j \in \mathcal{B}_t} \tilde{g}_j^{(i)} + \frac{1}{B} \xi^{(i)} \right\|^2 \\
&\overset{(iii)}{\le} f(w^{(0)}) + \frac{1}{2L} \|\nabla f(w^{(0)})\|^2 + L\eta^2 T^2 C^2 + \frac{L\eta^2 T}{B^2} \sum_{i=0}^{T-1} \|\xi^{(i)}\|^2 \\
&\overset{(iv)}{\le} f(w^{(0)}) + (f(w^{(0)}) - f^*) + L\eta^2 T^2 C^2 + \frac{L\eta^2 T}{B^2} \sum_{i=0}^{T-1} \|\xi^{(i)}\|^2 \\
&\overset{(v)}{\le} f(w^{(0)}) + (f(w^{(0)}) - f^*) + \frac{T^2 C^2}{L} + \underbrace{\frac{T}{LB^2} \sum_{i=0}^{T-1} \|\xi^{(i)}\|^2}_{Z_T},
\end{aligned}
\tag{16}
$$

where (i) is by the fact that $\langle a, b \rangle \le \frac{1}{2\lambda} \|a\|^2 + \frac{\lambda}{2} \|b\|^2 \ \forall \lambda > 0$, (ii) is by the convexity of $\| \cdot \|^2$, (iii) is because of $\|\tilde{g}_j^{(i)}\| \le C$ due to gradient clipping, (iv) is by the weak growth condition, (v) is by the assumption on learning rate ($\eta \le 1/L$).

By the definition of $\xi^{(i)}$, $Z_T$ follows from the sub-exponential distribution. By Lemma A.4, there exist some absolute constant $c_1 > 0$ such that

$$
\Pr\left[|Z_T| \ge t\right] \le 2\exp\left(-\frac{tc_1 LB^2}{T^2 C^2 d\sigma^2}\right).
$$

Combining with eq. (16), we have that

$$
\Pr\left[f(w_{\mathrm{priv}}) - f^* \ge 2(f(w^{(0)}) - f^*) + \frac{T^2 C^2}{L} + t\right] \le 2\exp\left(-\frac{tc_1 LB^2}{T^2 C^2 d\sigma^2}\right) \qquad \forall t \ge 0,
$$

which finishes the proof for eq. (14).

By the weak growth condition, we further have that

$$
\|\nabla f(w_{\mathrm{priv}})\|^2 \le 2L(f(w_{\mathrm{priv}}) - f^*)
$$

Therefore

$$
\Pr\left[\|\nabla f(w_{\mathrm{priv}})\|^2 \ge 4L(f(w^{(0)}) - f^*) + 2T^2 C^2 + 2Lt\right] \le 2\exp\left(-\frac{tc_1 LB^2}{T^2 C^2 d\sigma^2}\right) \quad \forall t \ge 0,
$$

which finishes the proof for eq. (15). $\qquad \square$

## APPENDIX C  MISSING EXPERIMENTS

### C.1  EXPERIMENTS ON SYNTHETIC DATA

The light-tail-noise assumption may not hold when training neural networks on real-world data. To keep consitent with our theory, we conduct experiment with synthetic data and artificial Gaussian noise. Our synthetic data has $10,000$ samples and each sample is generated from a 256-dimensional standard Gaussian distribution and a ground truth linear model. To simulate stochastic gradient with light-tailed noise, we perform linear regression and add standard Gaussian noise (light-tailed) for the true gradient. The experimental results are shown in Figure 4. The conclusion is the same as in

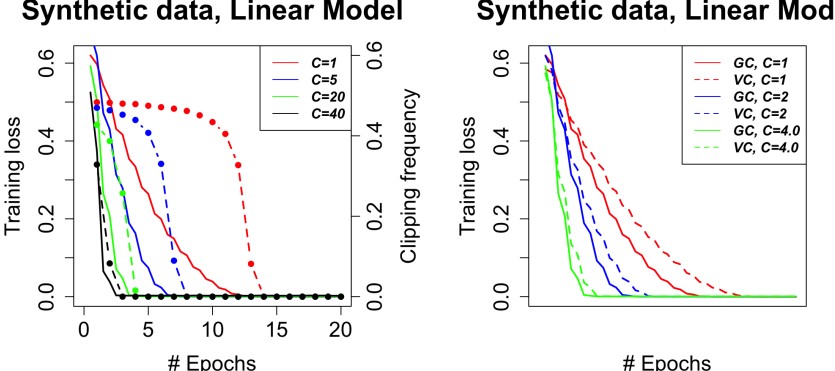

Figure 4: The evolution of clipping frequency and the comparison of gradient clipping (GC) and value clipping (VC).

Section 6: the evolution of clipping frequency has a decreasing trend and our value clipping technique can achieve similar accuracy as gradient clipping. The experiment in this part indicates that training neural networks on MNIST and CIFAR also enjoys light-tail-noise or the light-tail-noise assumption is removable.

## C.2 MORE EXPERIMENTS ON TESTING ACCURACY

In this section, we compare the testing accuracy of DP-SGD-GC (the baseline method) and DP-SGD-VC with different setup of $\epsilon$ and $\delta$. We set the noise level $\sigma \in \{0.5, 1, 2, 4, 8\}$ and confidence level $\delta \in \{10^{-3}, 10^{-4}, 10^{-5}\}$; each $(\sigma, \delta)$ pair decides a privacy parameter $\epsilon$, which can be calculated by the Opacus package. The result are shown in Figure 5.

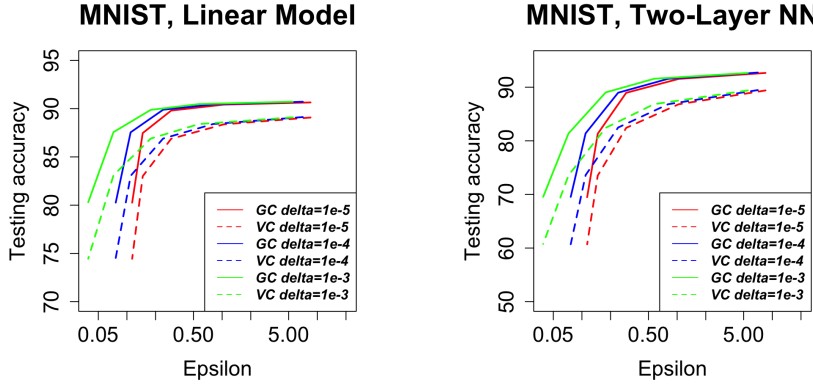

Figure 5: Testing accuracy of gradient clipping (GC) a value clipping (VC) with varying $\epsilon$ and $\delta$.

We can observe that the performance gap between WGC and VC grows as the privacy level $\epsilon$ becomes larger. For linear model, the testing accuracy gap is $\sim 2\%$ when $\epsilon \sim 10$ and the gap is $\sim 5\%$ when $\epsilon \sim 0.01$. For two-layer neural networks, the performance gap is larger, the testing accuracy gap is $\sim 3\%$ when $\epsilon \sim 10$ and the gap is about $\sim 9\%$ when $\epsilon \sim 0.01$. The experimental result should not be too surprising; imposing smaller $\epsilon$ implies adding more noise to the gradient, which will further increase the gradient norm and clipping frequency, and will eventually amplify the error created by VC.

## APPENDIX D MORE ON THE WEAK GROWTH CONDITION

We review some existing results and describe the weak growth condition for feed-forward neural networks with cross-entropy loss. First, we review the *generalized growth condition* (Fang et al.,

2021, Proposition 4.1). Consider the objective

$$\min_{w \in \mathbb{R}^d} f(w) := \frac{1}{n} \sum_{i=1}^{n} f_i(w) := \frac{1}{n} \sum_{i=1}^{n} \ell(h_i(w)), \tag{17}$$

where $\ell : \mathbb{R} \to \mathbb{R}_{\geq 0}$ is a *nonnegative* 1-dimensional loss function that is *convex*, 1-*smooth*, and satisfies $\inf \ell = 0$. The functions $h_i$'s are assumed to be $\beta$-Lipschitz continuous for some $\beta > 0$. Then $f_i$'s and $f$ are shown to satisfy the following weak growth condition.

**Lemma D.1** (Fang et al., 2021, Proposition 4.1). *For any $w \in \mathbb{R}^d$,*

$$\|\partial f_i(w)\|^2 \leq 2\beta^2 f_i(w) \qquad \forall i \in [n], \qquad and \qquad \|\partial f(w)\|^2 \leq 2\beta^2 f(w),$$

*where $\partial f(w)$ is the Clarke's generalized gradient (Clarke, 1981).*

We refer readers to Fang et al. (2021) for more details on the derivation of the above lemma. Note that $f_i$'s and $f$ in eq. (17) can be nonconvex, and the above lemma suggests that the weak growth condition holds for a certain class of nonconvex ERM.

### D.1 WEAK GROWTH CONDITION FOR FEED-FORWARD NEURAL NETWORKS WITH CROSS-ENTROPY LOSS

We give a simple extension of Lemma D.1 to feed-forward neural networks with cross-entropy loss.

We denote the number of classes as $K$. For simplicity, we consider a feed-forward neural network with fixed width $m$. We denote the parameter of a $H$-layer feed-forward neural network as $\mathbf{W} := (\mathbf{W}_1, \ldots, \mathbf{W}_H)$, where $\mathbf{W}_1 \in \mathbb{R}^{m \times d}, \mathbf{W}_H \in \mathbb{R}^{K \times m}$ and $\mathbf{W}_i \in \mathbb{R}^{m \times m}, i = 2, 3, \ldots, H-1$. We further denote $\|\mathbf{W}\|_F^2 = \sum_{i=1}^{H} \|\mathbf{W}_i\|_F^2$. With a little abuse of notation, we denote $\sigma$ as the ReLU activation, e.g., $\sigma(x) = (\max\{x_1, 0\}, \ldots, \max\{x_m, 0\})$. Note that neural networks with the ReLU activation is not differentiable everywhere on its domain. We define the "gradient" of a ReLU neural networks in the same way as in Allen-Zhu et al. (2019, Fact 2.6).

We consider a single training sample $x$ (which is enough for our purpose) and define the architecture of the $H$-layer feed-forward neural network as

$$
\begin{aligned}
h_0 &= x, \\
h_j &= \sigma(z_j), \ z_j = \mathbf{W}_j h_{j-1} \qquad \forall j \in [H-1], \\
\hat{y} &= \mathbf{W}_H h_{H-1},
\end{aligned}
$$

where $h_j$ is the hidden variables of the $j$-th layer, $\hat{y}$ is the prediction produced by the network.

Without loss of generality, we assume that the label for our training sample is the first class. Then by the definition of the cross-entropy loss, we have that

$$f(\mathbf{W}) \ = \ -\log\left(\frac{\exp(\hat{y}_1)}{\sum_{i=1}^{K} \exp(\hat{y}_i)}\right) \ = \ \log\left(1 + \sum_{i=2}^{K} \exp(\hat{y}_i - \hat{y}_1)\right).$$

Making the identification

$$\ell(\alpha) = \log(1 + \exp(\alpha)) \qquad \text{and} \qquad g(\hat{y}) = \log\left(\sum_{i=2}^{K} \exp(\hat{y}_i - \hat{y}_1)\right).$$

Then

$$f(\mathbf{W}) \ = \ \ell(g(\hat{y})).$$

It is obvious that $\ell$ satisfy the assumptions made by Lemma D.1. In order to apply Lemma D.1, all the remaining is to bound $\left\|\frac{\partial g(\hat{y})}{\partial \mathbf{W}}\right\|$. First, we notice that

$$
\begin{aligned}
\nabla_{\hat{y}} g(\hat{y}) &= \left[-1, \frac{\exp(\hat{y}_2 - \hat{y}_1)}{\sum_{i=2}^{K} \exp(\hat{y}_i - \hat{y}_1)}, \frac{\exp(\hat{y}_3 - \hat{y}_1)}{\sum_{i=2}^{K} \exp(\hat{y}_i - \hat{y}_1)}, \ldots, \frac{\exp(\hat{y}_K - \hat{y}_1)}{\sum_{i=2}^{K} \exp(\hat{y}_i - \hat{y}_1)}\right] \\
&\implies \|\nabla_{\hat{y}} g(\hat{y})\|_1 = 2 \\
&\implies \|\nabla_{\hat{y}} g(\hat{y})\|_2 \leq 2. \tag{18}
\end{aligned}
$$

Then we need to bound $\left\| \frac{\partial \hat{y}}{\partial \mathbf{W}} \right\|$. Directly applying Allen-Zhu et al. (2019, Fact 2.6) gives that

$$\left\| \frac{\partial \hat{y}}{\partial \mathbf{W}_i} \right\|_F \;\leq\; \|h_{i-1}\|_2 \prod_{j=i+1}^{H} \|\mathbf{W}_j\|_2 \;\leq\; \|x\|_2 \prod_{j=1,j\neq i}^{H} \|\mathbf{W}_j\|_2 \qquad (19)$$

for all $i \in [H]$.

Combining Lemma D.1, eq. (18) and eq. (19), we obtain the following weak growth condition

$$\|\partial f(\mathbf{W})\|_F^2 \;=\; \sum_{i=1}^{H} \|\partial_{\mathbf{W}_i} f(\mathbf{W})\|_F^2 \;\leq\; 8\|x\|_2^2 \sum_{i=1}^{H} \prod_{j=1,j\neq i}^{H} \|\mathbf{W}_j\|_2^2 f(\mathbf{W}).$$

Furthermore, by $\|\nabla_{\hat{y}} g(\hat{y})\|_2 \leq 2$ and Allen-Zhu et al. (2019, Fact 2.6). We also have

$$\|\partial f(\mathbf{W})\|_F^2 \;=\; \sum_{i=1}^{H} \|\partial_{\mathbf{W}_i} f(\mathbf{W})\|_F^2 \;\leq\; 4\|x\|_2^2 \sum_{i=1}^{H} \prod_{j=1,j\neq i}^{H} \|\mathbf{W}_j\|_2^2.$$

To sum up, we obtain

$$\|\partial f(\mathbf{W})\|_F^2 \;=\; \sum_{i=1}^{H} \|\partial_{\mathbf{W}_i} f(\mathbf{W})\|_F^2 \;\leq\; 4\|x\|_2^2 \sum_{i=1}^{H} \prod_{j=1,j\neq i}^{H} \|\mathbf{W}_j\|_2^2 \min\{1, 2f(\mathbf{W})\}.$$

To access the WGC-parameters, we need to compute $\|\mathbf{W}_j\|_2, j \in [H]$ in each iteration, the overhead can be amortized by increasing batch size.

