# OpenReview forum: "Improved Convergence of Differential Private SGD with Gradient Clipping"
_ICLR.cc/2023/Conference — ICLR 2023 poster_

### Official Review · Reviewer_aq32 · 2022-10-25

**Confidence:** 4
**Correctness:** 4
**Technical Novelty And Significance:** 3
**Empirical Novelty And Significance:** 3
**Recommendation:** 6

**Clarity, Quality, Novelty And Reproducibility:**

The paper is well-written and very clear. The authors provide proof sketch in the paper which improves readability.

**Strength And Weaknesses:**

The paper is well-written and the results are clear. In general the paper is good and provides some new insights.

Questions:

1)In the following paper the authors provide some lower bounds for DP optimization. In which they show that the boundedness of the diameter space is necessary. It should be discussed in the paper why the lower bound does not hold under the growth condition.

Bassily, Raef, Adam Smith, and Abhradeep Thakurta. "Private empirical risk minimization: Efficient algorithms and tight error bounds." 2014 IEEE 55th annual symposium on foundations of computer science. IEEE, 2014.

2) In Prop 4.2, the authors provide a "uniform" upper bound on the excess error of every iteration. This result and growth condition imply that we can choose a clipping norm such that with a high probability, we "never" clip the gradients. I think given that we are operating under this event, the standard convergence results for smooth convex function work. I want to understand how the analysis in the paper differs from this.

**Summary Of The Paper:**

This paper studies the problem of DP optimization in the convex setting. This paper considers the problem in the setting that the loss function  is smooth but not Lipschitz and the domain is not bounded. Instead the author assumes a growth condition which basically relates the gradient of a point to the optimal gap of that point.

Then, under these conditions the authors show that noisy SGD can be used to achieve an optimization gap of sqrt(d)/sqr(n)eps. Also, assuming light tail noise for the gradients the optimization gap can be improved to sqrt(d)/n eps.

Their analysis provide an approach to better clip the gradients, and using this result the authors propose a practical algorithm DP-SGD.

**Summary Of The Review:**

This paper considers the DP-Optimization problem. The main result is that under a natural growth condition, the lipschitzness and boundedness assumption can be removed and we can obtain a nearly optimal convergence rate.
I think this paper is interesting.

---

> ### Author Response · Authors · 2022-11-18
> **Response to Reviwer aq32**
>
> We would like to thank the reviewer for the helpful comments, please see our response to your questions in the following.
>
> Q: Relationship between our results and the lower bound from Bassily et al. [1].
>
> A: We would like to thank the reviewer for pointing out the connection between our work and the lower bounds in [1]. The lower bound established in [1] (their section 5.1 and 5.2) assumes a bounded domain, the functions that they use to illustrate the lower bounds (eq. (3) and eq. (5)) would be invalid if one allows the diameter of the domain to goes to infinity. In particular, the function in their eq. (3) will not be bounded below if $\|\mathcal{C}\| = \infty$, which would violated our assumption; and the function in their eq. (5) will be senseless if $\|\mathcal{C}\| = \infty$ (because the definition of the function relies on the boundary of $\mathcal{C}$). To sum up, the lower bounds developed in [1] only hold in the bounded domain and therefore do not conflict with our convergence result.
>
> There are some recent works that study the lower bound of DP-SGD in unconstrained case [2]. However, their lower bound requires restricting the objective function to be generalized linear models (GLM) and thus does not hold in our case. To our knowledge, the lower bound analysis of DP-SGD for unconstrained smooth problems is under-explored. In the updated manuscript, we emphasized the relationship between our utility bounds and existing lower bounds in the paragraph below our Remark 4.1.
>
>
> Q: About the proof outline?
>
> A: Thanks for your careful review of our technical analysis. Your comment on our proof template is almost the big idea behind our analysis. Conditioning on the event of clipping never happens (which holds with high probability), we directly apply the classic convergence analysis of SGD to obtain our Proposition 4.4. The only missing part in your comment is the translation between conditional expectation (Proposition 4.4) to expectation (Theorem 4.5), where our Lemma A.8 is the technical tool for this conversion.
>
>
>
> [1] Bassily, Raef, Adam Smith, and Abhradeep Thakurta. Private empirical risk minimization: Efficient algorithms and tight error bounds. FOCS 2014.
>
> [2] Shuang Song, Thomas Steinke, Om Thakkar, and Abhradeep Thakurta. Evading curse of dimension-
> ality in unconstrained private glms via private gradient descent. AISTATS 2021.

---

### Official Review · Reviewer_bKYj · 2022-10-25

**Confidence:** 3
**Clarity, Quality, Novelty And Reproducibility:** 1. The paper’s main body is well-orga…
**Correctness:** 4
**Technical Novelty And Significance:** 3
**Empirical Novelty And Significance:** 2
**Recommendation:** 8

**Strength And Weaknesses:**

## Strengths

- Removal of the restrictive bounded domain and Lipschitz continuity assumptions seems a novel and significant contribution.
- Proposed Value Clipping may be practically useful for certain settings. There is a potential for better scalability in comparison to standard gradient clipping.

## Weaknesses

- Numerical results are quite simple which is fine for me, as the main contributions are theoretical. Though they serve a well-illustrative purpose.

- Assumption 4.1 needs to be discussed in more detail. In my view, it is not enough to mention some not-very-recent references as the optimization field progressed quite a lot from that time. Namely, when does this assumption hold in practice? I see how it makes sense for a simple case of additive Gaussian noise for stochastic gradient estimates. But what are the assumptions on the distribution involved in the expectation? Does $i \sim \mathcal{U}[1, \dots, n]$ or other mini-batch samplings (better suitable from a differential privacy perspective) fit into this framework?


**Summary Of The Paper:**

This paper suggests convergence analysis of Differentially Private SGD with gradient clipping (DP-SGD-GC) in a smooth, unconstrained setting without bounded domain and Lipschitz continuity assumptions. Obtained results match the previously obtained bounds (for a more restrictive case) under an additional light-tail-noise assumption. In addition, the authors suggest a novel value-clipping technique and compare it to standard gradient clipping in a simple numerical study with neural networks on  MNIST and CIFAR-10 datasets.

**Summary Of The Review:**

Overall this is a good submission with a strong theoretical contribution and a good practical potential worth accepting to a conference.

**Minor comments**

I would like to ask the authors to adjust some of the citations by using the `\citet` LaTeX command not to duplicate text. Especially in the Related Work section.

Small typos on page 18 (top): lower indexes for $z_t \to z_i, u_t \to u_i$ and $Z_{T-1} \to Z_{t-1}$

---

> ### Author Response · Authors · 2022-11-18
> **Response to Reviewr bKYj**
>
> We would like to thank the reviewer for the helpful comments, please see our response to your questions in the following.
>
> Q: More explanation about Assumption 4.1?
>
> A: Thanks for your careful proofreading. To our knowledge, the light-tail-noise assumption or the bounded noise assumption are still the main-stream assumptions used in the optimization community, and we have included some recent important works that make use of these assumptions in the updated manuscript. Lipschitz loss and simple learning tasks such as linear regression or logistic regression with bounded objective value all satisfy Assumption 4.1. Training neural-networks is notoriously hard to analyze and may not satisfy this assumption in some extreme cases, for example when the neural-net is initialized badly. There are some recent works study the convergence of SGD with heavy-tail noise [1]. When the noise is heavy-tailed, we can only get a weaker utility bound (Theorem B.2). We have added a more detailed discussion about Assumption 4.1 on page 4.
>
> Q: Omitted steps in the proof.
>
> A: Thanks for your suggestion, we have included more proof details, espeically for the proof of Proposition 4.2 and 4.4.
>
>
> Q: Relation to the concurrent work [2].
>
> A: Thanks for pointing us to this interesting related work. We have added a detailed discussion to [2] in our related work section. The empirical discovery from [2] indeed contrasts the proof technique used in our theoretical analysis since their result indicates that a small clipping threshold may not hurt performance (though a small clipping threshold will create some challenges in theoretical analysis). Their findings suggest that our analysis may be further improvable; maybe it is possible to further exploit the problem structure and reduce the clipping threshold in our analysis. The phenomenon presented in [2] is interesting and worth future investigation.
>
>
> Q: Comment on the name of growth condition.
>
> A: Thanks for your suggestion. We have modified the name of growth condition to ``weak growth condition''. The concept of weak growth condition was first introduced in [3], and it is the same as our formulation in Definition 3.1.
>
> Q: More discussion on the limitation of value clipping.
>
> A: Thanks for your suggestion, we have added a slightly more detailed discussion on the limitation of value clipping in the paragraph above Section 7.
>
>
> Q: About the privacy-utility trade-off?
>
> A: Reviewer 49Nb also mentioned this point and we have added another experiment in the Appendix D, where we evaluate the performance gap between standard gradient clipping and our value clipping, please see more details for our response to Reviewer 49Nb.
>
> Q: Other minor comments.
>
> A: Thanks for your carefully proof-reading. We have modified some citation by using `\citet` and fixed the typos in the revised manuscript.
>
> [1] Gürbüzbalaban et al. The Heavy-Tail Phenomenon in SGD. ICML 2021.
>
> [1] Bu, Zhiqi, et al. "Automatic clipping: Differentially private deep learning made easier and stronger." arXiv preprint arXiv:2206.07136 (2022).
>
> [2] Sharan Vaswani, Francis Bach, and Mark Schmidt. Fast and faster convergence of SGD for over-parameterized models and an accelerated perceptron. AISTATS 2019.

---

### Official Review · Reviewer_49Nb · 2022-10-26

**Confidence:** 3
**Correctness:** 3
**Technical Novelty And Significance:** 3
**Empirical Novelty And Significance:** 3
**Recommendation:** 6

**Clarity, Quality, Novelty And Reproducibility:**

The paper is generally clearly written and all the math I checked is correct.

Here few more points about the writing, I believe these can be easily fixed:

- You mention in Alg. 1 that "Uniform randomly sample a batch $B_t$ with size $B$..." However, the results of (Abadi et al., 2016) holds for the Poisson subsampling, which you do not mention in the paper. To keep things rigorous, I think this should be added/corrected.

- In the first paragraph of 'NUMERICAL STUDY' Section you mention: "In Appendix, we also present some experimental results on synthetic data with light-tailed noise." Does this mean that the SGD noise in the MNIST/CIFAR-10 experiments is not sub Gaussian? I.e., the theory does not hold here? If so, I think would be better to state it more directly.

- Why is the main algorithm called DP-SGD-GC, when DP-SGD is a widely used acronym for the very same algorithm?

**Strength And Weaknesses:**

Pros:

- The paper is generally well written and easy to follow.
- The convergence analysis seems solid, it is impressive that same asymptotics are obtained as in (Bassily et al., 2014) for Lipschitz continuous loss functions.

Cons:

- The value clipping part is perhaps a bit weak part of the paper, e.g. the drop in test accuracy for MNIST is quite big (around 5%).
- Experiments could be improved: instead of showing all those training accuracy figures, would be more interesting to see how the value clipping behaves for different values of $\sigma$ and different values of $\varepsilon$.
- Some parts are a bit unclear, the presentation could be improved here and there. Let me elaborate:

How are actually the growth condition parameters $\beta_1$ and $\beta_2$ determined for the neural networks? I looked at Appendix E, and I only see bounds on the gradients, but there is nothing that explicitly says how to a priori bound $\beta_1$ and $\beta_2$. Could you elaborate on this and explicitly show how to determine them? How big is then the gap to the actual gradient norm, i.e. how loose are the bounds?

You write: "The value clipping step (line 4 of Algorithm 3) can be realized within one forward-backward propagation if the GC parameters are given in advance. Therefore DP-SGD-VC can be as fast as the vanilla SGD algorithm."

But are they given in advance in the experiments? Or do you compute them using the expressions of Appendix E at each iteration? As far as I see, to have rigorous DP guarantees, that is required.



**Summary Of The Paper:**

The paper provides a convergence analysis of DP-SGD and proposes a certain value-clipping method as an alternative to gradient clipping of DP-SGD. The convergence analysis leads to the same or better asymptotics than state-of-the-art results with slightly weaker assumptions, and the value-clipping leads to faster compute with similar experimental results however requires certain growth condition to be satisfied (for the loss function).

The convergence analysis relies on the assumption that the SGD noise of the gradients is 'light-tailed' meaning it is sub.Gaussian and also on the RDP analysis of the subsampled Gaussian mechanisms provided by Abadi et al. (2016) which connects the level of DP noise and epsilons. This way convergence bounds w.r.t. epsilons and deltas can be provided.

**Summary Of The Review:**

All in all, I feel this is a borderline case and I hope the authors can clarify some of my concerns.

---

> ### Author Response · Authors · 2022-11-18
> **Response to Reviewer 49Nb**
>
> We would like to thank the reviewer for the helpful comments, please see our response to your questions in the following.
>
> Q: Experiments can be improved.
>
> A: Thanks for your insightful suggestion on our experiments. Following your suggestion, we have added a new subsection in Appendix D (we place it in appendix due to space limitation). The new experiment presents the performance of standard gradient clipping (GC) and our value clipping (VC) under different $\\epsilon$ and $\\delta$. We find that the performance gap between GC and VC grows as $\\epsilon$ decreases. Please see more of our detailed discussion in Appendix D.2.
>
> Q: How are the growth condition parameters $\\beta_1$ and $\\beta_2$ decided in our experiments?
>
> A: Thanks for your careful proofreading. Following the derivation in Appendix E, we set $\\beta_1 = 4 || x ||_2^2  \\sum\_{i=1}\^H  \\prod\_{j=1, j \\neq i}\^H \\| \\mathbf{W}_j \\|_2^2 \min \\{ 1, 2 f( \\mathbf{W} )\\} $ and $\\beta_2 = 0$ in our experiments,
> where $\\beta_2$ is known as a priori and $\\beta_1$ needs to be recomputed every iteration during training since the upper bound of gradient norm depends on the spectral norm of each layer of NN, which is varying during trainig. We have include more details on the computation of $(\\beta_1, \\beta_2)$ in the Section 6.2 in the updated manuscript.
>
> Q: Is value clipping giving advance in the experiments?
>
> A: Our implementation of VC indeed relies on our derivation in Appendix E. In each iteration of DP-SGD-VC, the algorithm first calculates the upper bound of the gradient norm (which depends on the spectral norm of each layer) for each sample in the mini-batch, then it imposes different weights for different samples and performs the backward step. Overall, the implementation of VC can be realized within one forward-backward step and gives clear advantages in terms of training efficiency; VC is much faster than Micro-batching (standard gradient clipping) and GC-Opacus. Please see the results in Table 2 for more details on the training time of different methods (the runtime recorded in Table 2 includes the overhead of computing $\\beta_1$).
>
>
> Q: Some points about writing and presentation.
>
> A: Thanks for pointing out these presentation issues. We have updated the manuscript following your suggestions. (i) We have corrected the sampling method in SGD to Poisson sampling. (ii) We include experiments with synthetic data to keep consistent with the light-tail-noise assumption. For the training of neural nets on MNIST and CIFAR, it is possible that the gradient noise can be heavy-tailed in some extreme cases (for example, if the NN has a bad initialization). However, the conclusion obtained from synthetic data and MNIST/CIFAR are the same, which implies that the gradient noise from MNIST/CIFAR does not conflict with our theory. (iii) In the literature of DP-optimization, DP-SGD with and without gradient clipping are both referred as DP-SGD. We call our algorithm DP-SGD-GC because we want to distinguish DP-SGD with/without gradient clipping in a clear way.

---

### Public Comment · ~Xiaodong_Yang7 · 2022-11-14
**Expect for clarifications on some technical points**

Introducing the *growth condition* and *sub-Gaussian gradient noise* into the theoretical studies of private empirical minimization, this manuscript also made contributions by proposing a well-motivate variant--value clipping.

However, I got confused when reading the detailed proofs of this manuscript, and got stuck especially in the **proof of Proposition 4.4**, presented in Appendix C.3, page 20-21. Two **non-trials gaps** exist:

$\bullet$ **Circular Argument in applying Proposition 4.3.** In more details, Propositions 4.2&4.3 sequentially establish high-prob upper bounds on the function value difference $\nabla f(w^{(t)})-f^\ast$, but the series $w^{(t)}$ should come from a **non-clipping method, Algorithm 2.** In the first half of the proof of Proposition 4.4, the authors **directly apply Proposition 4.3 to the series output by Algorithm 1 (which involves gradient clipping),** obtaining a high-prob upper bound on the gradient norm $\nabla f_i(w^{(t)})$, thus concluding clipping doesn't happen. This process involves some circular argument.

$\bullet$ **Ignoring Bias Term when adopting Lemma A.11** This lemma is taken from Theorem 2.1 (Ghadimi & Lan, 2013, https://arxiv.org/abs/1309.5549). The authors seem to miss a very important condition in this theorem: **gradient estimates need to be unbiased,** $\mathbb{E}[\nabla f_\xi(w)]=\nabla f(w)$. But the second half of the proof of Proposition 4.4 applies this theorem to Algorithm 1, in which there might be conditional bias for the gradient estimate. Specifically, event $\mathcal{E}$ involves the randomness of drawing mini-batches $B_t\subset[T]$, so $B_t$ is not independent of event $\mathcal{E}$, leading to possibly $\mathbb{E}[\frac{1}{B}\sum_{i\in\mathcal{B}_t}\nabla f_i(w^{(t)})|\mathcal{E}]\neq\nabla f(w^{(t)})$. **The authors seem to forget to checking the unbiased condition.**

Besides these two gaps, I also spot other statements to be refined. To name a few, the authors directly use Theorem 3.4 (Theorem 1 in (Abadi et al, 2016, https://arxiv.org/abs/1607.00133)) to prove the privacy guarantee. But in fact, (Abadi et al, 2016) employs Poisson subsampling to draw the mini-batches, different from the uniform sub-sampling employed in Algorithm 1 here. So the privacy accounting should be refined a little more.

Specifically, I think it would be nice for the authors to cite compare more recent results in private non-convex erm, like https://arxiv.org/pdf/2206.07136.pdf, https://arxiv.org/pdf/2206.13033.pdf, and the references therein. A comprehensive comparison on used assumptions and conclusions would help everyone better identity the contributions of this manuscript.

---

> ### Author Response · Authors · 2022-11-18
> **Thanks for your interest and some clarifications on some technical points**
>
> Dear Xiaodong:
>
> The authors are pleased to receive additional comments from the community. Indeed, this is a very active area of research with great potentials/impacts in real-world industrial applications. We hope more PhD students and researchers will be attracted to this topic and grow the community.
>
> Thank you for your specific questions regarding the proof. The updated manuscript has added some more detailed derivations: we include more details on the connection between the conditional bounds for SGD and DP-SGD (the first part of the proof of Proposition 4.4 is based on non-clipped SGD and does not involve circular argument); and on the convergence of SGD conditioned on a event that happens with high probability. Please see Page 21 of the updated manuscript and let us know if your questions are answered satisfactorily.
>
> We have also added the related work mentioned in your comment. Again, we thank you for your interest in our work.
>
> Sincerely,
>
> Authors.

---

### Decision · Program_Chairs · 2023-01-20

**Decision:**

Accept: poster

**Justification For Why Not Higher Score:**

The experiments could be improved and the practical benefits of the new clipping techniques seem limited.

**Justification For Why Not Lower Score:**

The paper expands our understanding of private SGD by removing restrictive assumptions that do not hold in important scenarios.

**Metareview: Summary, Strengths And Weaknesses:**

The main contribution of the paper is a new analysis of private SGD with clipping that removes restrictive assumptions for smooth convex functions, namely the bounded domain assumption and the assumption that the function is globally Lipschitz continuous. If the noise from the subsampled gradient are light-tailed (sub-Gaussian), the guarantees obtained are comparable to those established in prior work under the additional assumptions. The paper is primarily theoretical and focuses on improving the theoretical understanding of private SGD. On the practical side, the paper introduces a new clipping technique for functions that satisfy the weak growth condition, which is a relaxation of smoothness, that can be implemented efficiently.

The reviewers agree that the theoretical contribution of the paper is relevant and interesting, and it is strong enough to merit acceptance. The reviewers felt that the experiments could be improved and that the practical benefits of the new clipping techniques were limited.


**Note From Pc:**

if the above contains the word "oral" or "spotlight" please see: "oral" presentation means -> notable-top-5% and "spotlight" means -> notable-top-25%. As stated in our emails, we are disassociating presentation type from AC recommendations